# Structural colour enhanced microfluidics

Detao Qin [1,2], Andrew H. Gibbons[1,2], Masateru M. Ito [1,2✉], Sangamithirai Subramanian Parimalam [1], Handong Jiang[1,2], H. Enis Karahan [1,2], Behnam Ghalei [1,2], Daisuke Yamaguchi[1,2], Ganesh N. Pandian[1,2] & Easan Sivaniah [1,2✉]

Advances in microfluidic technology towards flexibility, transparency, functionality, wearability, scale reduction or complexity enhancement are currently limited by choices in materials and assembly methods. Organized microfibrillation is a method for optically printing well-defined porosity into thin polymer films with ultrahigh resolution. Here we demonstrate this method to create self-enclosed microfluidic devices with a few simple steps, in a number of flexible and transparent formats. Structural colour, a property of organized microfibrillation, becomes an intrinsic feature of these microfluidic devices, enabling in-situ sensing capability. Since the system fluid dynamics are dependent on the internal pore size, capillary flow is shown to become characterized by structural colour, while independent of channel dimension, irrespective of whether devices are printed at the centimetre or micrometre scale. Moreover, the capability of generating and combining different internal porosities enables the OM microfluidics to be used for pore-size based applications, as demonstrated by separation of biomolecular mixtures.

[1] Institute for Integrated Cell-Material Sciences (iCeMS), Kyoto University of Advanced Study, Kyoto University, 606-8501 Kyoto, Japan. [2] Department of Molecular Engineering, Kyoto University, 616-8510 Kyoto, Japan. ✉email: mito@icems.kyoto-u.ac.jp; esivaniah@icems.kyoto-u.ac.jp

Current-day microfluidics has found a strong foothold in applications from DNA sequencing[1,2], to point-of-care diagnostics[3–5], to organ-on-a-chip models for drug testing[6–8]. However, the manufacturing of such microfluidic chips, especially those with high resolution, narrow channel width, is dependent on lithography to create a master template that is either etched into a rigid material such as glass, silicon, or ceramic[9,10], or soft lithographic techniques whereby a mold is first created, and then replicated into a curable polymer, most commonly polydimethylsiloxane (PDMS)[11–13]. Inevitably, the process of creating a complex channel, and then sealing its open surface against another material to create an enclosed channel, is a significant practical obstacle to advances in microfluidic devices. A more recent development is using paper or fabric microfluidics, where a porous fabric is coated in pattern with hydrophobic chemistry[14,15]. However, though cheap, flexible, and practical in remote locations, such methods have limitations in the resolution of the devices and the control of internal pore sizes[16,17], and hence the down-sizing capability.

Photolithography itself is generally considered in a positive or negative sense. Positive photoresists become degraded under the light so that the exposed region of the polymer film is washed away by a developer solution to create structure; negative photoresists work in the opposite sense, and both are used to make channels[18]. In either case, the upper surface of the channel is left open and must be sealed to become an enclosed microfluidic channel (Fig. 1a). Recently, organized microfibrillation (OM) was reported as a process that uses photolithographic principles to create porous sub-structure self-enclosed within the polymer film[19]. The substructure itself is unique in that it is composed of periodic non-porous layers separated by porous microfibril layers. The dimensions of the non-porous layers can be tuned by the wavelength of crosslinking light, and this has been demonstrated

for wavelengths between 250 and 405 nm. Equally, the porosity of the porous layers is capable of adjustment. This alternating arrangement of porous and non-porous layers gives rise to Bragg-reflections of the incident light, thus creating structural colour effects.

As the internal porous sub-structure of these OM pixels are connected in plane, the OM process itself could be used to print complex self-enclosed channels for microfluidic flow (Fig. 1b). In this paper, we demonstrate the ease of such a microfluidic platform printing technology, examine the dynamics of capillary flow within such channels, and explore the interaction of flow behaviour with their intrinsic structural colour properties. Finally, we show the capability of generating different internal porosity in a single miniature OM device, and further use this functionality for pore-size based separation of biomolecules.

## Results

**Fabrication of OM microfluidic devices.** OM microfluidic channels were fabricated by exposing photosensitive polymer films on a reflecting surface to a monochromatic light, while using a shadow mask for making the desired patterns. The films are then developed in a weak solvent to create self-enclosed channels with an internal porous structure. An advantage of the OM process is that the whole microfluidic pattern can be produced at the same time with a shadow mask. Low-cost masks can be prepared with ordinary overhead transparency (OHP) sheets, and conventional printers are suitable for this task down to millimetre scale details. Archetypal microfluidic motifs, such as cross-junction, hexagon-web, and zigzag channels, were created in this way to make the macrochannel on a number of substrates (Fig. 1c–e, Supplementary Tables 1 and 2). Due to its mirror-like surface, silicon wafer is a suitable substrate to make OM

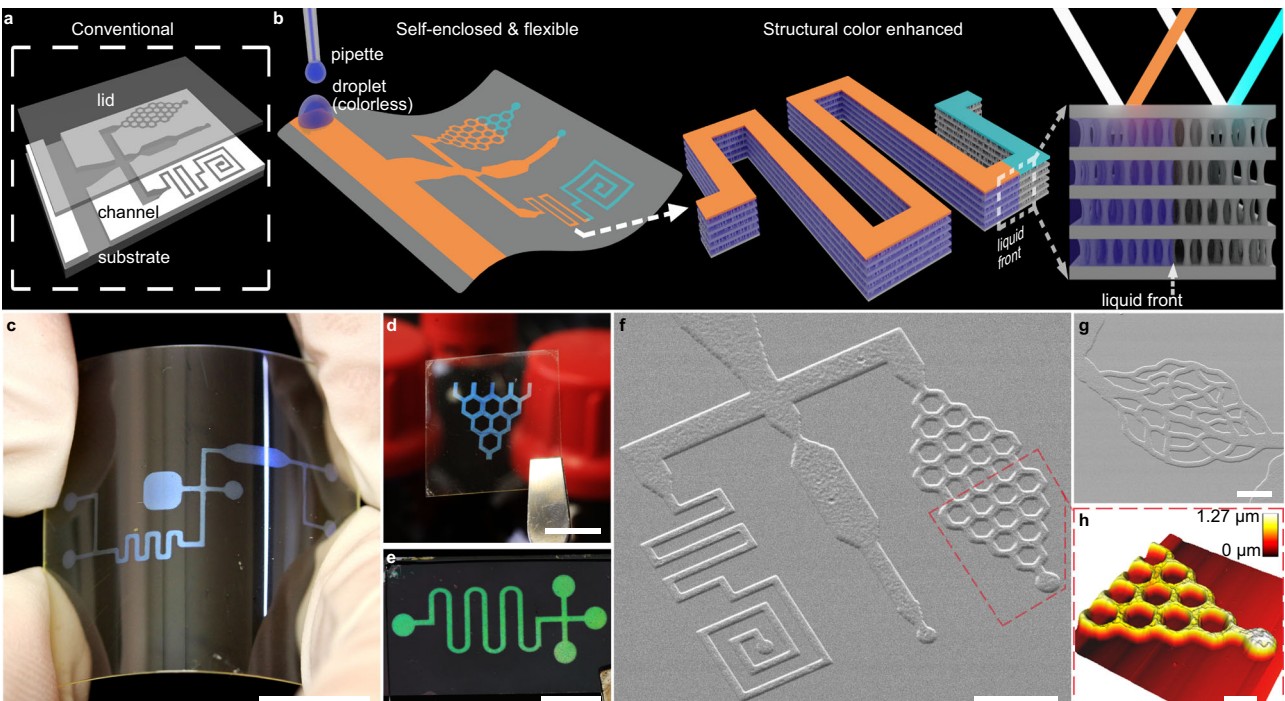

**Fig. 1 Self-enclosed flexible microfluidics with intrinsic structural colour. a** Conventional lithographic techniques for microfluidic devices require a sealing step, as shown by the schematic. **b** Illustration for microfluidics made by organized microfibrillation (OM). Flow-induced colour change (as depicted from cyan to orange) results from liquid infiltration changing the optical property of the periodic OM structure. **c–e** Macro lens photos of OM channel printed on different substrates: (**c**) poly(ethylene terephthalate) (PET) sheet, (**d**) cover glass, and (**e**) silicon wafer. **f, g** SEM images for OM microfluidics with (**f**) a complex channel and (**g**) a capillary-like pattern, respectively. **h**, AFM image for the same OM microchannel in (**f**). Scale bar, (**c–e**) 1 cm, (**f**) 50 μm, (**g**) 10 μm, (**h**) 20 μm.

microfluidics on (Fig. 1e). Moreover, OM channels can be made on transparent and flexible substrates by placing the film face down on a reflecting surface at the photocrosslinking step (Fig. 1c, d). The fabricated OM microfluidic device has visible-light structural colour due to its periodic internal structure, where the structural colour can be tuned by illumination light wavelength and other experimental conditions. A Bragg peak is observed for the resulting microfluidic channels in their reflectance spectra (Supplementary Fig. 1).

The resolution of the microfluidics is limited by the detail of the shadow mask. To achieve ultra-high resolution channel designs, a micro-LED illumination instrument was used for photocrosslinking. The micro-LED allows pixel-level control of light illumination, down to 1 micron. To demonstrate this capability, we created highly detailed microfluidic flow channels. Two designs are demonstrated: a capillary-like structure; and a collection of miniaturized microfluidic features, including a hexagonal lattice, width variation, spiral, and zigzag. Here we can show OM microchannels with feature sizes down to 5 μm in width. The out-of-plane expansion of the porous layers allows the channels to be observed with a scanning electron microscope (SEM), despite the channels being self-enclosed within the polymer film (Fig. 1f, g). Topography investigations by both an atomic force microscope (Fig. 1h, Supplementary Fig. 2a–c) and a digital microscope (Supplementary Fig. 2d–f) confirm the out-of-plane expansion feature of OM channels.

While the extrinsic channel width can be on the order of micrometres to millimetres, the internal pore sizes remain in the submicron length scale. The internal porous structure of the OM channels provides a large surface area, which facilitates liquid flow by capillary action. For small devices, capillary action is a convenient mechanism to drive liquid flow[20–22]. Capillary flow speed depends on material interactions and pore geometry[23–26]. For the microfluidics made by polystyrene (PS), the channels are wettable by a number of common liquids such as alcohols and alkanes (Supplementary Fig. 3). Therefore, these liquids are readily driven by capillary forces to initiate the flow inside the channel. Highly detailed visualization of the liquid flow is achieved using fluorescent dyes with a confocal laser scanning microscope (CLSM). Liquid is introduced into the channels by placing a droplet on the surface of the film. The liquids permeated through the channel surface and flow was further induced by capillary force. The confocal micrographs confirm that the internal porous regions are well connected, and the liquids can flow through the patterned channel as designed (Fig. 2a, b). This spontaneous permeation mechanism is convenient for introducing liquids into small channels and is a proof of concept for using these microfluidic channels for contact-based testing, for example, sticking the device to the skin.

Moreover, liquid mixing was demonstrated in OM channels. Two solutions based on ethanol, dosed with red and green fluorescent dyes respectively, were mixed within a T-junction channel and monitored under CLSM. Laminar flow of the streams was observed with diffusive mixing at the boundary of the solutions (Fig. 2c). In Fig. 2, both silicon wafer (Fig. 2a, b) and cover glass (Fig. 2c, d) were used as the channel substrate. Figure 2d shows an example of printing microchannels on cover glass (channel width: down to 8 μm) using the micro-LED instrument.

**Flow behaviour and sensing capability of OM microfluidics**. A unique feature of the OM microfluidic platform compared to other methods is the dependence of flow speed on porosity rather than pattern geometry. The pore size, which determines the flow speed, is intrinsic to the internal porous layer in OM channels.

Therefore, fluidic property becomes independent from extrinsic channel width, thus OM channels can have arbitrary shapes without impacting the flow behaviour. This concept is demonstrated by observing liquid flow in microfluidic channels of varying width but with the same internal porous structure produced by micro-LED (Fig. 3a). The flow parameter $dL^2/dt$ ($L$, observed flow distance; $t$, time), which describes how fast the liquid spreads in the capillary microfluidics, is shown to be independent of the channel width (Fig. 3b). Such flow behaviour is distinct from conventional hollow channels. When comparing the flow parameter to a model of conventional hollow channel: $dL^2/dt$ would be halved for a 20-μm channel compared to a 40-μm channel (Supplementary Discussion).

In OM channels, the size of internal pores not only impacts fluidic properties but also correlates to interlayer spacing and structural colour, thus making the structural colour and flow speed the coupled properties. We investigated the relationship between microfluidics of different colour (Bragg peak position) and their flow dynamics. Using $n$-hexadecane as a model liquid, the square of the flow distance ($L^2$) was found to be proportional to time (Fig. 3c, Supplementary Fig. 4). The behaviour of the microfluidic systems follows the Lucas-Washburn model of capillary flow[27]. We further characterized the flow in OM microfluidics by a modified Lucas-Washburn model[28–30], as shown in the below equation (See Supplementary Discussion for the derivation):

$$L^2 = \frac{\alpha^3 r_a \gamma \cos\theta}{2\mu_e \tau^2} t \tag{1}$$

where $\alpha$ is pore geometry factor, $r_a$ is average effective pore radius, $\tau$ is tortuosity, $\gamma$ is surface tension, $\theta$ is contact angle, and $\mu_e$ is effective viscosity of the liquid in the channel.

The results demonstrate that our microfluidic devices can visualize the fluidic properties of the porous channel as a colorimetric value: A red-shifted structural colour indicates larger interlayer spacing, hence faster fluid flow through that structure. The coupling of microfluidic properties with structural colour holds true regardless of channel size (Fig. 3d, Supplementary Table 3). Such a coupling effect was tested over two orders of magnitude from 5 mm (straight-line channels) down to 5 μm (various designs printed by micro-LED) and was shown to be true at all tested length scales (Supplementary Figs. 5 and 6). The takeaway of this result is that the periodic porous structure of our microfluidics is independent of the channel design, while only depending on the internal structure itself. Since the extrinsic channel geometries define the fluid flow properties in conventional microfluidic chip design, the OM platform offers a higher degree of freedom for pattern designs that could be realized and optimized for diverse applications.

Different fluid properties within OM channels were examined. Two solvent families, alkanes and alcohols, were studied to test the effect of liquid viscosity. For each family, pure and mixed solvents were used (Supplementary Tables 4 and 5). The results from both families show an inverse correlation between the flow parameter ($dL^2/dt$) and liquid viscosity (Fig. 3e, Supplementary Fig. 7), further confirming the Lucas-Washburn flow behaviour. As a result, OM microfluidics is able to make the distinction between the two families based on viscosity sensing.

The displacement of air with liquid by infiltration produces a change of refractive index (RI), hence liquid flow in OM channel can be observed by naked eye (Fig. 4a, Supplementary Movie 1). The Bragg peak in the reflectance spectrum is present after the periodic structure is filled with liquid. This provides a basis for in-situ sensing[31,32]. Using different solvents from the alcohol family, the Bragg peak position of the microfluidic channel shows a linear

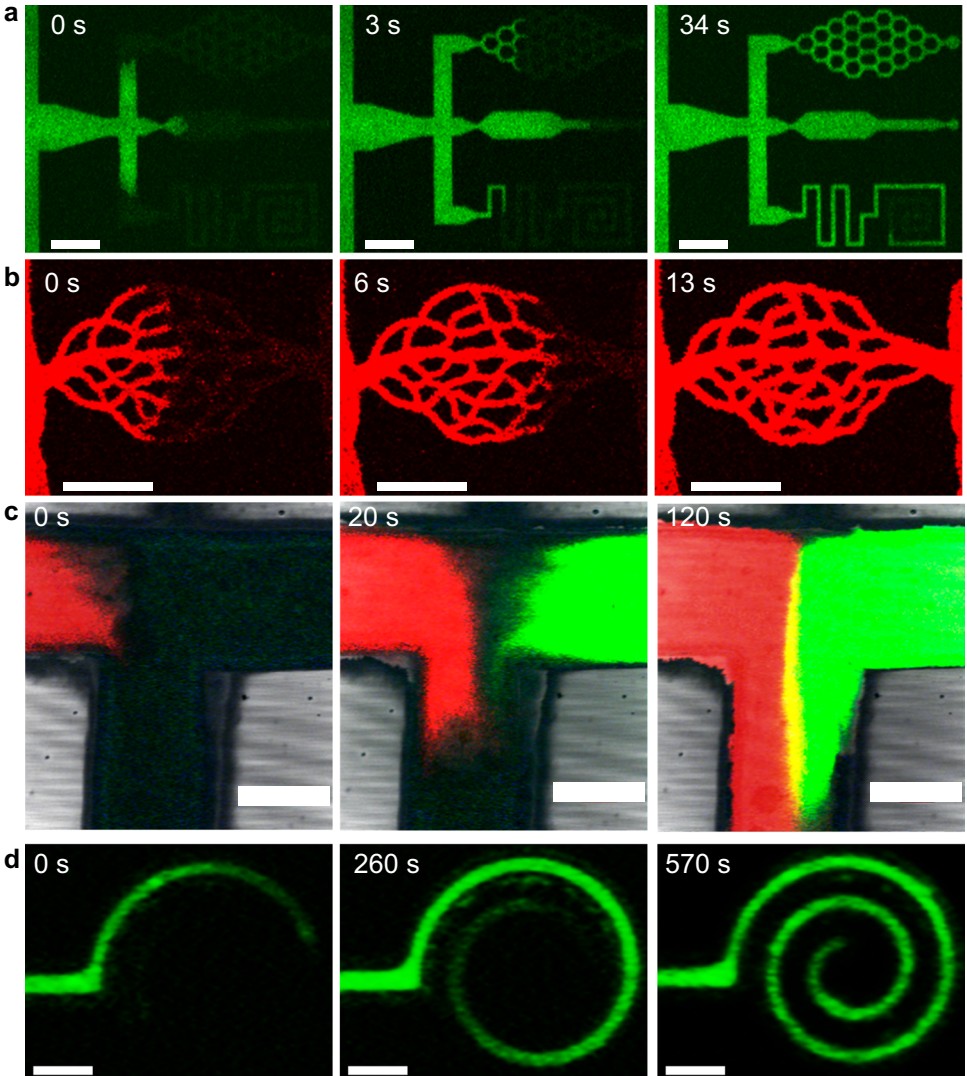

**Fig. 2 Confocal laser scanning microscope photos for liquid flow in OM channels.** Liquid flow through (**a**) a complex channel and (**b**) a capillary-like pattern, respectively. **c** Two stream mixing at a T-junction channel. **d** Aqueous solution flow through a spiral pattern. In (**a**–**c**), the liquid is ethanol/water (9/1, v/v), and the fluorescent dyes are ATTO 495 (green) and ATTO 610 (red). In (**d**), the liquid is an acrylamide aqueous solution (5 wt%) and the fluorescent solute is a green fluorescent protein. Substrate: (**a**, **b**), silicon wafer, (**c**, **d**) cover glass. Scale bar, (**a**, **b**) 100 µm, (**c**) 0.5 mm, (**d**) 20 µm.

response to the refractive index of the added solvent (Fig. 4b). Using a different polymer material, poly(methyl methacrylate), for the fluid channel causes the response of the liquid to change (Fig. 4c), which is attributed to the different surface energy interaction between the polymer and liquids. Apart from Bragg peak position, the peak reflectance value was also found to vary with solvent refractive index, showing a negative correlation (Fig. 4d). Moreover, the ability to detect solute concentration based on refractive index was also demonstrated for OM channels (Supplementary Fig. 8).

Figure 4e exhibits a proto-type microfluidic device sticking on human finger, showing in principle the wearability of OM technology. This particular sample was made by polycarbonate, a durable polymer compatible with the OM process. OM films made by polycarbonate were transferred onto aluminium foil (60-µm thickness) to study the effect of bending on the structural colour. The in situ spectrum shows negligible colour change when the curvature of film is ≤0.1 mm$^{-1}$. As the curvature further increases to 0.5 mm$^{-1}$, the Bragg peak shift of OM film remains in the small range of 1–4 nm. The spectrum remains almost unchanged after releasing the bending and returning the film to silicon wafer (see Supplementary Fig. 9 for details).

**Porosity manipulation and biomolecular separations**. Up to this point, the microfluidic flow has been demonstrated mainly with alcohols and alkanes, due to their affinity to polystyrene. However, water is the most commonly used liquid for microfluidics as it is a solvent for numerous biomolecules. Achieving aqueous flow in OM microfluidics would extend its scope to biomedical applications. The challenge is that the polystyrene OM is not hydrophilic (water contact angle greater than 90°, Supplementary Fig. 3a). To address this problem, we identified an additive that can improve the affinity between the polystyrene OM channels and water-based systems. Acrylamide, a commodity chemical frequently used in biomedical research and industry, including as a material for electrophoresis gels, was found to be effective in enabling aqueous flow in OM channels. The water/acrylamide system has a contact angle (time = 0 s) of 63° on polystyrene OM surface, reducing to 30° within 1 min (Supplementary Fig. 3b). The improved affinity allows for aqueous flow through the OM channels (Fig. 2d, flow marker is a green fluorescent protein). This capability enables OM microfluidics to be used in biomedically relevant tests which use an aqueous environment.

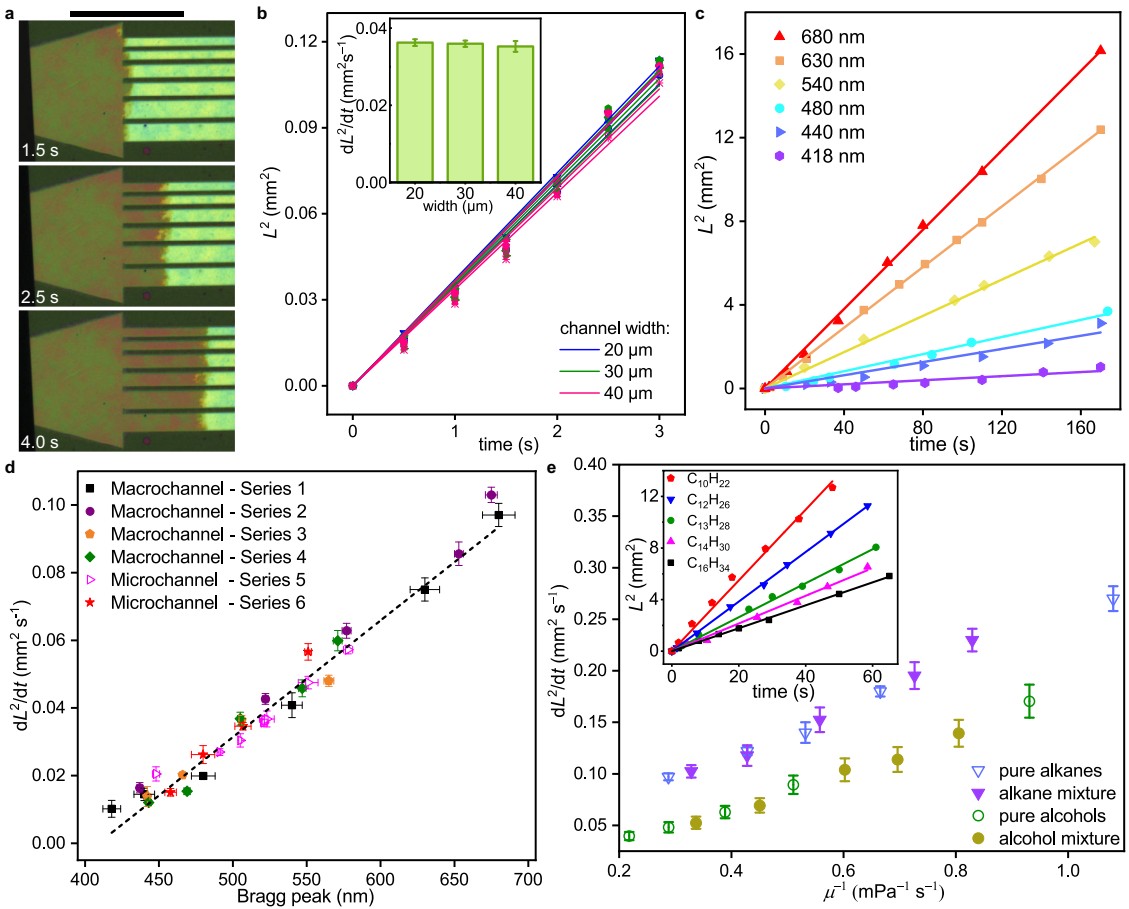

**Fig. 3 Flow dynamics of OM channels. a** Optical microscope photos showing flow in OM channel is independent of extrinsic channel width. Scale bar, 200 µm. **b** Flow behaviour in microbars, $L^2$ versus $t$. Inset shows statistics of $dL^2/dt$ versus channel width. Error bars are standard deviation (S.D.) from 4 measurements. **c** $L^2$ versus $t$ for OM channels with different Bragg peaks. **d** Correlation between fluidic property and structural colour as shown by $dL^2/dt$ versus Bragg peak location. Optical microscope photos for microchannels are shown in Supplementary Fig. 5. Series 1, 2 are flow measurements from 3-mm-width channels; series 3, 4 are flow measurements from 5-mm-width channels. Series 5, 6 are flow measurements from micron-sized channels (features from 5 to 40 microns). The channels, prepared under different photoinitiator and illumination conditions, are specified in Supplementary Table 3. Error bars are S.D. across 3 measurements. Liquid is $n$-hexadecane for (**a–d**). **e** The dependence of $dL^2/dt$ on liquid viscosity. In detail, pure alkane refers to $n$-decane, $n$-dodecane, $n$-tridecane, $n$-tetradecane, and $n$-hexadecane; alkane mixture refers to the binary mixture of $n$-decane and $n$-hexadecane with different mole fractions; pure alcohol refers to ethanol, $n$-propanol, $n$-butanol, $n$-pentanol, and $n$-hexanol; alcohol mixture refers to the binary mixture of ethanol and $n$-hexanol with different mole fractions. The calculation of mixture viscosity is presented in Supplementary Tables 4 and 5. Inset shows the $L^2$ versus $t$ for pure alkanes. The $L^2$ versus $t$ for alkane mixture, pure alcohols, and alcohol mixture is presented in Supplementary Fig. 7. Error bars are S.D. across 3 samples.

The advantage of OM technology for biomedical applications is the convenience for making varying porosity at submicron scale in a single miniature device, while the differing porosities are correlated to its structural colours and can be tailored for specific purposes. Through controlling the crosslinking energy at the local regions of the print pattern, the internal porosity of the OM microchannels can be varied and combined. As a demonstration of this concept, one can supply a high energy dose to the main channel and lower doses to the side branches (further details in Supplementary Fig. 10). With this design, OM microchannels with pore size variation were printed on cover glass (Fig. 5a, b, Supplementary Fig. 11). The microscope photos in Fig. 5c–e clearly show that the main channels (high energy dosage) have a more reddish structural colour than the side branches (low energy dosage). Moreover, SEM cross-sections show that the channel with high energy dosage contains larger internal pores (Fig. 5f–h, Supplementary Fig. 12), confirming the positive correlation between structural colour and internal pore size. In addition, in Fig. 5 the microchannels of pore-varying features were printed

on glass substrate using the micro-LED instrument. Note that the channels were printed on glass to facilitate viewing by inverted microscopes.

The capillary flow of biomolecules in such microchannels was performed and proof-of-concept studies were conducted to demonstrate the pore-size based separation functionality. Polysaccharides and proteins of various molecular weights were selected to prepare the aqueous solutions of mixing solutes. The aqueous solution was introduced to the open cross-section of the OM main channel and could flow into the separation region with branch of smaller pore size. The visualization of biomolecular separation was conducted by an inverted CLSM, and the biomolecule solutes can be differentiated by their fluorescence (see details in the Methods section and Supplementary Table 6). Figure 6a shows the separation of 3-kDa dextran from 70-kDa dextran in a Y-type channel with different porosity in each branch. In both branches, 3-kDa dextran permeates faster than the 70-kDa one. 70-kDa dextran permeates faster in the large-pore branch compared to the small-pore branch. This indicates

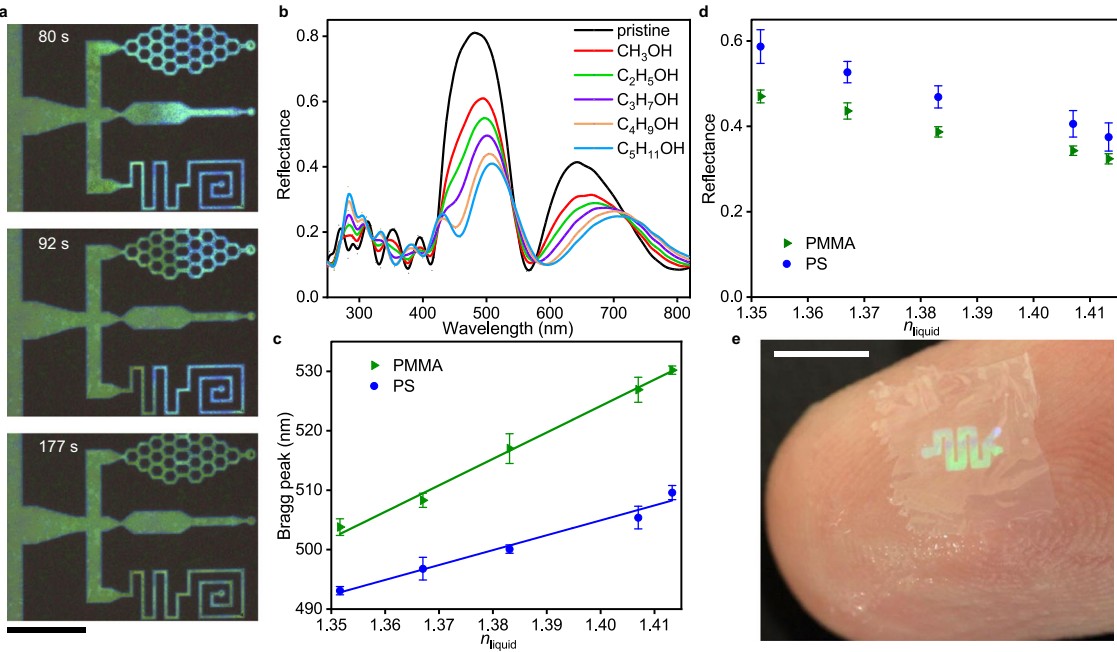

**Fig. 4 In situ sensing by OM microfluidics. a** Colour change observed by optical microscope as liquid (*n*-hexadecane) flowing through OM microchannels. The photos are extracted from Supplementary Movie 1. **b** Reflectance spectrum for different alcohols flowing in OM channels. **c** Bragg peak location and (**d**) peak reflectance versus the refractive index of the inflow liquid. **e** A macro lens photo showing an OM microfluidics sticking on human finger. Scale bar, (**a**) 100 μm, (**e**) 0.5 cm. Error bars in (**c**) and (**d**) are S.D. across 3 samples.

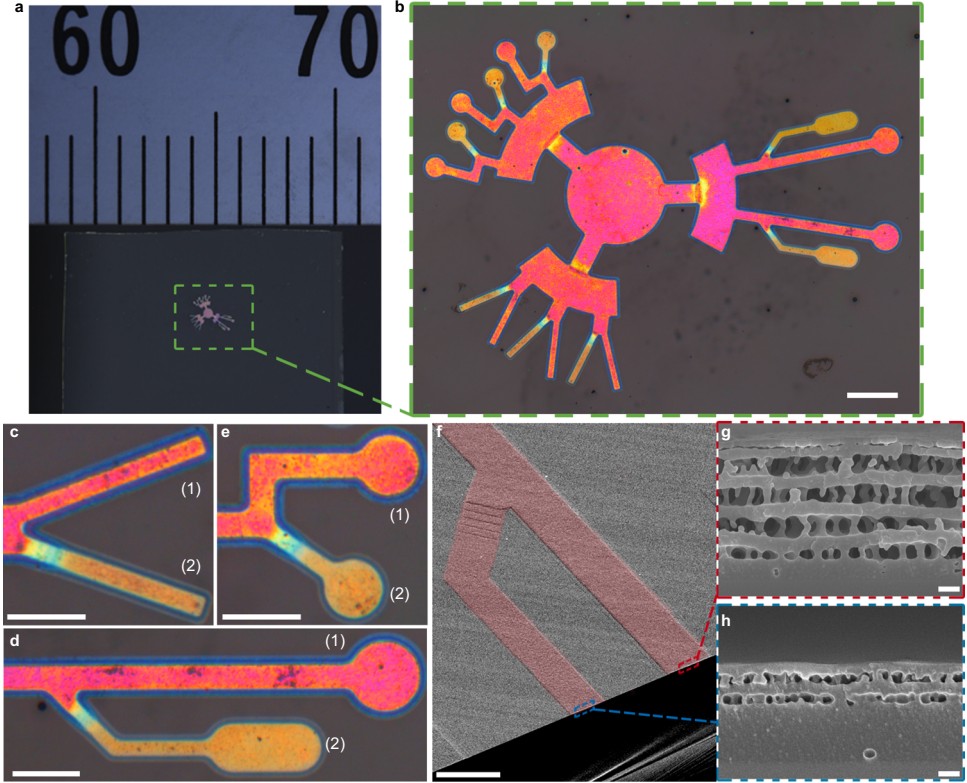

**Fig. 5 Manipulation of internal pore size with crosslinking energy. a** Photo of OM microchannel made on cover glass. Variation of branch colour is accomplished by changing energy dosage according to the design as specified in Supplementary Fig. 10: the main channel adopted a high energy dosage while the side branch adopted a low energy dosage. **b** Microscope photo of the same OM microchannel in (**a**). **c–e** The corresponding zoom-in microscope photos, where the main channels (labelled as (1), energy dosage: 600 mJ/cm$^2$) show a more reddish structural colour compared to the side branches (labelled as (2), energy dosage: 300 mJ/cm$^2$). **f** An SEM image for the OM microchannel in (**d**). Red colour is added to highlight the channel. The original image is shown in Supplementary Fig. 12. **g** The zoom-in SEM cross-section for the main channel in (**f**). **h** The zoom-in SEM cross-section for the side branch in (**f**). Scale bar, (**b**), 200 μm, (**c–e**) 100 μm, (**f**), 50 μm, (**g**, **h**), 200 nm.

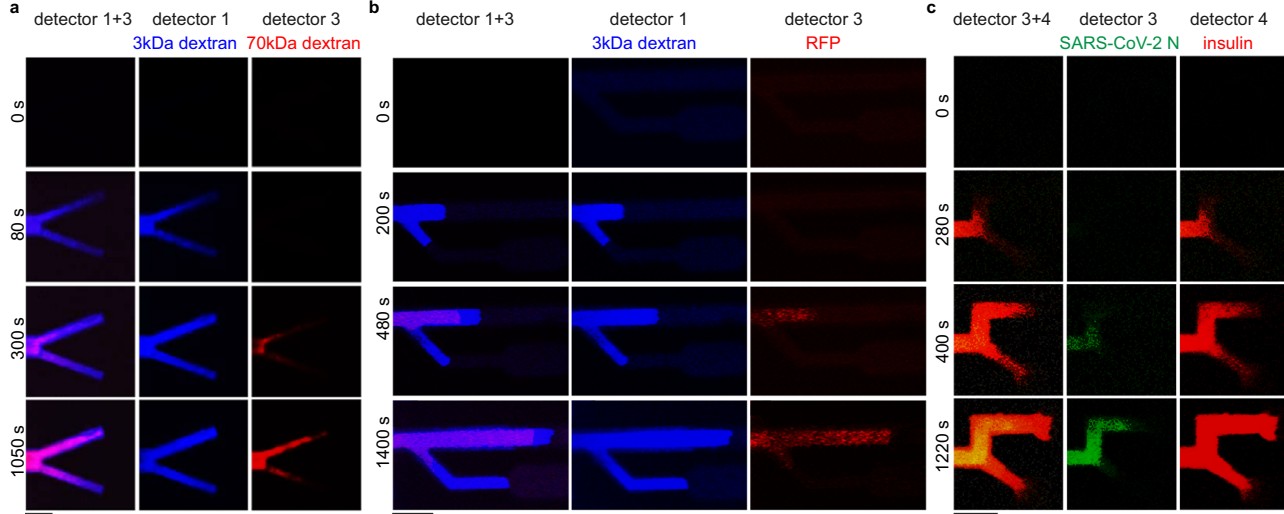

**Fig. 6 Biomolecular separations by OM microchannels.** Imaging was conducted by an inverted confocal laser scanning microscope (CLSM, Nikon A1R). **a** Separation of two polysaccharides of different molecular weights, where the blue and red colours represent 3-kDa (Cascade Blue labelled) and 70-kDa (Texas Red labelled) fluorescent dextran, respectively. **b** Separation of polysaccharide from protein, where the blue and red colours represent a fluorescent dextran (3 kDa) and a red fluorescent protein (RFP, 27 kDa), respectively. **c** Separation of two proteins of different molecular weight, where the green and red colours represent SARS-CoV-2 nucleocapsid protein (55 kDa, pre-stained with the dye SYPRO Orange) and fluorescent insulin (Alexa Fluor 680 labelled, 6 kDa), respectively. These OM microchannels were made on cover glass using a micro-LED instrument ($\lambda_i = 405$ nm). Microchannels were printed according to the design in Supplementary Fig. 10 with different energy dosages in the main channel and the side branch. The microscope photos of these OM channels are shown in Fig. 5c–e, respectively. Further experimental details are provided in Supplementary Table 6. Scale bar, (**a–c**), 100 μm.

70–kDa dextran encounters more resistance in the small-pore branch. These results demonstrate that the OM microfluidic channels can distinguish biomolecules by controlling the internal pore sizes, and this can be achieved in submillimeter length scale with a micrometre-scale printing resolution. Stronger separation ability can be observed for the separation of a low molecular-weight dextran (3 kDa) from a large molecular-weight protein (red fluorescent protein, 27 kDa), see Fig. 6b.

Protein separation was first examined with Bovine Serum Albumin (BSA, 66 kDa) and insulin (6 kDa), which are often used as model proteins in biomedical assays[33–35]. The results show BSA is more difficult to permeate through the small-pore branch, and hence the small-pore branch achieves a purified product of insulin (Supplementary Fig. 13). There is a recent surge in the evidences linking Diabetes mellitus with the pathophysiology of COVID-19[36,37]. As a demonstration of this contemporary relevance, Severe acute respiratory syndrome coronavirus 2 (SARS-CoV-2) nucleocapsid protein (55 kDa) and insulin were tested for the separation. SARS-CoV-2 nucleocapsid protein shows different permeability in the separation channels (Fig. 6c). We found protein-protein and protein-dextran mixtures achieved a better separation compared to dextran-dextran mixture. Likely this is due to dextran having a molecular weight distribution while each protein has a specific molecular weight. With these demonstrations, we can present OM microfluidics as a new platform relevant for biomedical research and applications.

## Discussion

Generating an advanced platform technology for the creation of microfluidic devices represents a game-changing opportunity for the future of wearable, analytical and sensing technologies. We have demonstrated the creation of microfluidic devices using a novel photolithography technique generally applicable to commodity polymers such as polystyrene, poly(methyl methacrylate), as well as polycarbonate. We have also demonstrated the capillary flow of several classes of liquids including aqueous solutions in OM microfluidic channels. The ease of combining different

internal porosity in a single miniature OM device enables pore-size sieving functions and applications. This approach removes some of the steps typically required in conventional microfluidic devices, allowing quick development of microfluidic designs on multiple substrate types. Our microfluidic devices provide an opportunity to develop applications for filtration, skin contact technology, and in situ sensing and analysis.

## Methods

**Materials.** Commercial grade polymers polystyrene (PS, weight-averaged molecular weight $M_w$, 35 kDa), poly(methyl methacrylate) (PMMA, $M_w$, 120 kDa), and bisphenol A polycarbonate (PC, $M_w$, 45 kDa) were purchased from Sigma-Aldrich. Photoinitiator 4,4′-bis-(diethylamino)-benzophenone (BDABP) was purchased from Sigma-Aldrich. Photoinitiator 9,10-phenanthrenequinone (PQ) was purchased from Tokyo Chemical Industry. The solvents, including chloroform, dichloromethane, and acetic acid were obtained from Nacalai Tesque. Alkanes and alcohols were purchased from Tokyo Chemical Industry.

Acrylamide (≥99%), sodium dodecyl sulfate (SDS, ≥ 99%), and phosphate-buffered saline (PBS, 10× concentrate) were purchased from Sigma-Aldrich. Fluorescent dyes ATTO 495 (Ex/Em: 495/527 nm) and ATTO 610 (Ex/Em: 615/634 nm) were purchased from Sigma-Aldrich. The green fluorescent protein (rAcGFP1, Ex/Em: 475/505 nm) was obtained from Takara Bio. The red fluorescent protein (eqFP611, Ex/Em: 559/611 nm) was purchased from Abcam. Fluorescent dextrans, 3 kDa (Cascade Blue labelled, Ex/Em: 400/420 nm, Invitrogen™) and 70 kDa (Texas Red labelled, Ex/Em: 595/615 nm, Invitrogen™), were obtained from Thermo Fisher Scientific. The fluorescent insulin (Alexa Fluor 680 labelled, Ex/Em: 679/702 nm) was obtained from Nanocs Inc. The dye SYPRO Orange (Ex/Em: 470/570 nm, 5000× concentrate, Invitrogen™) was purchased from Thermo Fisher Scientific. Bovine Serum Albumin (BSA, B9000S) was purchased from New England Biolabs. SARS-CoV-2 Nucleocapsid protein (aa1-419) was obtained from Thermo Fisher Scientific. Deionized (DI) water (18 MΩ cm) was obtained from a Milli-Q ultrapure water system (Integral 5, Millipore). Polydimethylsiloxane (PDMS) sheet was prepared using Sylgard 184 kit from Dow Corning. Typical substrates for film casting in this study were cover glass (Muto Pure Chemicals), poly(ethylene terephthalate) (PET) sheet (Lumirror T60, 250 μm thickness, Toray Industries), and silicon wafer (<100 >, N-type, single side polished, Matsuzaki Seisakusyo, FC-100). Aluminium foil (LUXAL UV, 60-μm thickness, Toyo Aluminium K.K.) was used as the substrate for the mechanical bending tests.

**OM microfluidics fabrication.** OM channels were fabricated in both micro and macro scales in this study. The term "microchannel" refers to channel width

printed at micrometre scale while the term "macrochannel" refers to channel width printed at millimetre scale.

The fabrication sequentially executes film casting, photocrosslinking, and development, irrespective of the scale for fabrication. Prior to film casting, polymers were purified according to a previous study[19]. Briefly, the solution of commodity polymers was filtered through a 0.2-μm-pore PTFE syringe filter (mdi Membrane Technologies), followed by mixing with DI water and phase separation for four cycles, and then reprecipitated in methanol. After filtering through a Whatman filter paper (grade 41), the collection was dried in a vaccum oven at 50 °C overnight to obtain the purified polymer. Photoinitiator and purified polymer were dissolved in the casting solvent (either chloroform or dichloromethane) to prepare the solution. The solution was sonicated for 30 min, filtered through a PTFE syringe filter (0.2-μm-pore, mdi Membrane Technologies), and spun-cast on a clean substrate (MS-A100 spin coater, Mikasa). More details about film casting are provided in Supplementary Table 1.

For OM macrochannel, the cast film was illuminated by LED bulbs (Thorlabs, $\lambda_i$: 300–405 nm) using the custom crosslinking ovens (cubic enclosure: 10 cm width × 10 cm height × 18 cm length, 6 bulbs in 3 rows × 2 columns hung up at ceiling centre, kept at room temperature by a USB fan on top). Microfluidic pattern designs were printed onto an overhead transparency (OHP) sheet (FX-471P, folex IMAGING) as the shadow mask, which was put in-between the light source and polymer film during the crosslinking. Film placement depends on whether the substrate is transparent. Polymer films cast on a reflective substrate (e.g., silicon wafer) were placed flat, face-up towards the light source. In contrast, polymer films on a transparent substrate (e.g., PET sheet or cover glass) were placed face-down on a mirror surface (e.g., silicon wafer). After photocrosslinking, the film was immersed in a developer solution (acetic acid for PS and PC, acetic acid/water = 4/3, v/v for PMMA) at room temperature (20 ± 2 °C). The immersion was immediately finished upon the completion of colour changes in the illuminated region. The developed sample was dried using an air blower.

For OM microchannel, film casting and development were the same with macrochannel as mentioned above, while a micro-LED instrument (Maskless D-light DL-1000 GS/KCH, NanoSystem Solutions, $\lambda_i$: 405 nm) was employed for the photocrosslinking instead. The micro-LED instrument is combined with a digital mirror device (1024 × 768 micromirrors), with an individual micromirror controlling the on/off status of light projection onto a 1-μm square region. More details about channel preparation are provided in Supplementary Table 2.

**Flow dynamics measurement**. Capillary flow dynamics were studied systematically for both macro and micro OM channels. The channels of different structural colour (Bragg peak) were prepared through controlling illumination light wavelength and other experimental conditions (see Supplementary Table 3 for more details on channel fabrication). OM channel samples were cut by a diamond knife for exposing the porous cross-section, upon which liquid was introduced to the channel for dynamics measurement. For OM microchannel, flow dynamics was observed by an optical microscope (Axioscope A1 MAT, Carl Zeiss). For OM macrochannel, flow dynamics was observed by either this microscope or a digital single-lens reflex (DSLR) camera.

The experimental setup using a DSLR camera to measure flow dynamics in OM macrochannel was schematically illustrated in Supplementary Fig. 4. In particular, straight-line OM macrochannels were printed using stainless steel stencil as the shadow mask. The channels were suspended vertically from the above. Liquid bath was slowly lifted up by a lab scissor jack until one open channel end is immersed. Liquid flow in OM macrochannels was observed by a DSLR camera (EOS Kiss X5) equipped with a macro lens (EFS 60 mm, Canon).

The liquid for flow dynamics study is *n*-hexadecane unless otherwise stated. For all the measurements, flow distance ($L$) was determined by the photo analysis software *ImageJ* (version 1.52p) using its built-in ruler tool. Three samples were investigated for reporting the statistics on arithmetic mean value and error bar.

**Confocal laser scanning microscope (CLSM)**. Depending on substrate opacity, either an upright or inverted microscope was employed to record the fluorescence microscope images.

In particular, the upright CLSM (FV-1000, Olympus) was used for OM channels printed on a reflective substrate (e.g., silicon wafer), while the inverted CLSM (A1R Eclipse Ti, Nikon) was used for OM channels printed on a transparent substrate (e.g., cover glass). The confocal microscopes are equipped with four lasers (410, 489, 561, 638 nm) and four associated detectors. Each detector records the emission spectrum of a different wavelength range (detector 1: 425–475 nm, detector 2: 500–550 nm, detector 3: 570–620 nm, detector 4: 663–738 nm). A particular detector with the strongest signal was selected for each fluorescent dye or biomolecule. Biomolecules with distinct fluorescence colours were selected for making mixture solutions. The differentiation of biomolecules in the mixture can be achieved by using multiple detectors simultaneously. Additionally, a differential interference contrast (DIC) detector can be used to observe the shape of OM channel. More details about CLSM experiments are provided in Supplementary Table 6.

**Characterizations**. The structures of OM microfluidics were observed by a scanning electron microscope (JEOL JSM-7500F). Prior to SEM observation, a 5-nm-thick osmium layer was deposited on the sample by sputter coating (Osmium Plasma Coater OPC60A, Filgen). OM channel topography was characterized by both an atomic force microscope (NanoWizard III, JPK instruments) and a digital microscope (VHX-7000, Keyence). Macro lens photos were taken by a DSLR camera (EOS Kiss X5, EFS 60 mm macro lens, Canon). Microscope photos were taken by an upright optical microscope (Axioscope A1 MAT, Carl Zeiss). The DSLR camera and optical microscope were white balanced using an 18% neutral grey card prior to photo taking. The reflectance spectra of OM channels were measured by a spectrometer (MCPD-3700, Otsuka Electronics) with a 210–820 nm light source (MC-2530, Otsuka Electronics, Japan). All reflectance spectra were normalized with the substrate (e.g., bare silicon wafer) as the reference. Film thickness was determined by the optical analysis software of the same manufacturer using Cauchy equation[38]. Contact angles were measured by an optical goniometric instrument (DSA25S, KRÜSS, GmbH) using the sessile drop technique.

## Data availability

The authors declare that all the relevant data within this paper and its Supplementary Information are available from the corresponding authors upon request.

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

## Acknowledgements

This work was supported by the Japan Society for the Promotion of Science (JSPS) KAKENHI (20H00390), JST-PRESTO (JPMJPR1417), and the JST-START program (JPMJST2015). The central facilities are supported by the World Premier International Research Initiative (WPI), MEXT, Japan. We thank the Nanohub at Kyoto University for access to their facility and instruments. We thank the Analysis Centre at iCeMS, KUIAS, Kyoto University for access to their SEM and confocal microscopes. We thank Prof. Motomu Tanaka and Dr. Ryo Suzuki for access to their confocal microscope, as well as providing training and assistance. M.M.I. acknowledges JSPS KAKENHI (20K15342). A.G. and H.E.K. acknowledge the JSPS International Fellowship programme. S.S.P is supported by JSPS KAKENHI (JP19H003349 and 21H0475) to G.N.P.

## Author contributions

D.Q., M.M.I., and E.S. conceived the idea and designed the study. D.Q. conducted the major experiments, analyzed the data, and wrote the manuscript. M.M.I. and E.S. supervised the project. S.S.P. advised on biomolecular experiments. A.G., M.M.I., H.J., H.E.K., B.G., D.Y., and G.N.P. supported in data analysis. All authors discussed the results and contributed to manuscript refinement.

## Competing interests

The authors declare no competing interests.
