## [Peer Review File · Nature Communications]

REVIEWER COMMENTS

Reviewer #1 (Remarks to the Author):

This paper reports the application of the OM technique pioneered by Sivaniah's group to microfluidics. It makes sense -- since OM generates a porous layer using photolithography, it follows that it should be possible to generate porosities that are useful for capillary flow.

I enjoyed the cleverness of the paper, the paper is well written, and I think it would make a great Lab-on-a-chip-style paper, but for Nature Communications, I was expecting something that went beyond the demonstration of flow in the OM-patterned structures. It feels like we are reading about a very expensive way of making capillary flows on plastic, when one could use paper instead. The authors need to think about a "killer app" for which OM is optimally suited and generate one more figure with it in order to be suitable for publication in Nature Comm.

Reviewer #2 (Remarks to the Author):

The current manuscript extends the work presented by the authors in their 2019 Nature paper (Nature, 2019, 570, 363), with periodic porous structures being printed on flexible substrates to create self-enclosed channels for capillary flow. In this current work, both fluid velocity and structural color variations within microfluidic channels are found to be related to the viscosity and refractive index of the solvents used, thus providing a basis for in-situ sensing. Additionally, the current studies report the use of simpler, cheaper and higher resolution fabrication methods for fabricating capillary microfluidic chips. The work is generally well-presented and the results nice. That said, the sensing aspects of the work appear most interesting to me, but at the moment only simple proof-of-principle studies are shown. This is disappointing and it would be far more useful for the authors to show sensing operation using real world samples. Accordingly, and since the underlying concepts of the technical approach have been published previously, I would recommend that the authors provide a revised manuscript that provides much more relevant sensing data. This should address real-world samples and benchmark performance with respect to existing sensing methods.

Specific Comments

1.

The authors are successful in fabricating structures on flexible substrates, but do not explain key features or benefits. For example, in Fig. 1c and Fig. 4c, it seems likely that the bending of the substrate influences the spectra (the color is not uniform across the whole substrate).

2.

The authors present a method for sensing the viscosity of a liquid by monitoring the velocity of flow in the channels (Fig. 3e). The tests are carried out using a pure hydrocarbon. How will the measurement be effected if the sample is a combination of multiple chemicals? Is the capillary flow velocity only dependent on the viscosity? Will it still show the same trend with other types of solvents?

3.

Similarly, in the refractive index testing, the authors choose alcohol to prove the relationship. What happens when the solvent is changed?

4.

The authors characterize the refractive index via the Bragg peak (Fig. 4b). But the peak intensity varies more than the Bragg peak. Why would one not choose the peak intensity as the indicator?

Overall response

We greatly appreciate both reviewers' complimentary and constructive comments that helped improve the quality of our manuscript. As head of a research group that emphasizes practical implementation of our research findings through entrepreneurship, I appreciate that the comments highlighted the practical aspects of our work. We took these comments as a renewed call-to-action.

Over the past six months, we have focused on addressing the crucial points emphasized by the reviewers (Reviewer #1: "killer app"; Reviewer #2: "real world samples" and additional results on sensing phenomenon).

We are pleased to announce that we have achieved two major breakthroughs through substantial effort:

- (1) Submicron-scale manipulation and combination of different porosities in a single miniature device, which enables pore-size based applications.
- (2) In-channel water flow, which open new vistas of opportunities for OM microfluidics in biomedically relevant research and applications, such as the separation of various biomolecules (*e.g.*, polysaccharides, proteins including BSA, insulin, SARS-CoV-2 nucleocapsid protein).

I would like to emphasize the second breakthrough. We also felt that our first submission lacked the "killer-app" status due to lack of biologically relevant demonstrations. Microfluidics will have a broad range of application, including the oil-and-gas and fine-chemical synthesis sectors. However, the dominant application sector is in health sensing, medical diagnosis and wearables. Despite the trial of various surfactants in the preceding 4 years, we had not succeeded to achieve sufficient capillary flow of waterborne biomolecules and biomarkers into our OM microchannels.

We are grateful for the reviewers' feedback that led us to make major experimental additions to the paper in the intervening 6 months. We began a collaboration with a biomedical group who suggested a number of commonly used materials in biological assays. These additions suitably enhanced the hydrophilicity of OM microchannels. They were also able to recommend a number of real-world biomolecules that might have interest to the microfluidic biological assay community.

In the revised version of our manuscript, we included demonstrations of these two breakthroughs along with two new figures (Figs. 5 and 6).

In addition, we made another noteworthy improvement: the printing of OM microchannels on transparent substrates *e.g.* cover glass (Our previous publication *Nature* 363, 579 introduced micro-printing on silicon wafer only). This achievement facilitated the observation of OM microfluidic devices by inverted microscopes, which are commonly used for microfluidic flow tests, as shown in the new figures.

With these new findings and other impactful results on structural colour enabled sensing capability, we are confident that we managed to show the versatility of the OM microfluidics platform, potentially opening up new horizons for the concept of microfluidics.

Please find our replies for Reviewer #1 and Reviewer #2 in a point-by-point format below. For enhanced clarity, we colour-coded the text change highlights in (i) yellow for Figures 5 and 6 (in response to both Reviewers), and (ii) turquoise for responding to additional comments of Reviewer #2, and (iii) light gray for the other common changes and general text improvements.

Best regards,
Prof. Easan Sivaniah

Response (in blue) to comments (in black) from Reviewer #1 (text change highlight in yellow)

This paper reports the application of the OM technique pioneered by Sivaniah's group to microfluidics. It makes sense -- since OM generates a porous layer using photolithography, it follows that it should be possible to generate porosities that are useful for capillary flow.

I enjoyed the cleverness of the paper, the paper is well written, and I think it would make a great Lab-on-a-chip-style paper, but for Nature Communications, I was expecting something that went beyond the demonstration of flow in the OM-patterned structures. It feels like we are reading about a very expensive way of making capillary flows on plastic, when one could use paper instead. The authors need to think about a "killer app" for which OM is optimally suited and generate one more figure with it in order to be suitable for publication in Nature Comm.

We appreciate the reviewer's comment on finding an optimally suited "killer app" of OM technology, which motivated us to further explore this aspect of our technology for most of this year.

After making a series of breakthroughs, we are now able to show that OM microfluidics is able to generate differing internal porosities at submicron scale in a single miniature device (Fig. 5). By making use of this functionality, we could also demonstrate pore-size based separation of biomolecules (Fig. 6).

We think that these two achievements demonstrate an advantage of OM microfluidics over paper-based platforms. The convenience of pore-size variation and combination at submicron scale in a well-defined way is a unique feature of OM microfluidic technology, which cannot be achieved by paper microfluidics due to their limitations in device resolution and internal pore size control^{1,2}.

Plus, combined with the original finding of structural colour enabled sensing capabilities, now we can present OM microfluidics as a new and versatile platform relevant for a wide range of research topics and hopefully practical applications.

Overall, we hope that the reviewer will acknowledge the fact that the technology we present is not limited to a single "killer app" but brings a new perspective to device design for microfluidics. The revisions made in the main text provide further insights, which are as follows:

Main text, page 8-11, line 172-231:

"Porosity manipulation and biomolecular separations. Up to this point, the microfluidic flow has been demonstrated mainly with alcohols and alkanes, due to their affinity to polystyrene. However, water is the most commonly used liquid for microfluidics as it is a solvent for numerous biomolecules. Achieving aqueous flow in OM microfluidics would open new vistas of opportunities in biomedical applications. The challenge is that the polystyrene OM is not hydrophilic (water contact angle greater than 90°, Supplementary Fig. 3a). To address this problem, we identified an additive that can improve the affinity between the polystyrene OM channels and water-based systems. Acrylamide, a commodity chemical frequently used in biomedical research and industry, including as a material for electrophoresis gels, was found to be effective in enabling aqueous flow in OM channels. The water/acrylamide system has a contact angle (time = 0 s) of 63° on polystyrene OM surface, reducing to 30° within one minute (Supplementary Fig. 3b). The improved affinity allows for aqueous flow through the OM channels (Fig. 2d, flow marker is a green fluorescent protein). This capability enables OM microfluidics to be used in biomedically relevant tests which use an aqueous environment.

The advantage of OM technology for biomedical applications is the convenience for making varying porosity at submicron scale in a single miniature device, while the differing porosities are correlated to its

structural colours and can be tailored for specific purposes. Through controlling the crosslinking energy at the local regions of the print pattern, the internal porosity of the OM microchannels can be varied and combined. As a demonstration of this concept, one can supply a high energy dose to the main channel and lower doses to the side branches (further details in Supplementary Fig. 10). With this design, OM microchannels with pore size variation were printed on cover glass (Fig. 5a, b, Supplementary Fig. 11). The microscope photos in Figs. 5 c-e clearly show that the main channels (high energy dosage) have a more reddish structural colour than the side branches (low energy dosage). Moreover, SEM cross-sections show that the channel with high energy dosage contains larger internal pores (Fig. 5f-h, Supplementary Fig. 12), confirming the positive correlation between structural colour and internal pore size. In addition, in Fig. 5 the microchannels of pore-varying features were printed on glass substrate using the micro-LED instrument. Note that the channels were printed on glass to facilitate viewing by inverted microscopes.

The capillary flow of biomolecules in such microchannels was performed and proof-of-concept studies were conducted to demonstrate the pore-size based separation functionality. Polysaccharides and proteins of various molecular weights were selected to prepare the aqueous solutions of mixing solutes. The aqueous solution was introduced to the open cross-section of the OM main channel and could flow into the separation region with branch of smaller pore size. The visualization of biomolecules separation was conducted by an inverted CLSM, and the biomolecule solutes can be differentiated by their fluorescence (see details in the Methods section and Supplementary Table 6). Fig. 6a shows the separation of 3-kDa dextran from 70-kDa dextran in a Y-type channel with different porosity in each branch. In both branches, 3-kDa dextran permeates faster than the 70-kDa one. 70-kDa dextran permeates faster in the large-pore branch compared to the small-pore branch. This indicates 70-kDa dextran encounters more resistance in the small-pore branch. These results demonstrate that the OM microfluidic channels can distinguish biomolecules by controlling the internal pore sizes, and this can be achieved in submillimeter length scale with a micrometer-scale printing resolution. Stronger separation ability can be observed for the separation of a low molecular-weight dextran (3 kDa) from a large molecular-weight protein (red fluorescent protein, 27 kDa), see Fig. 6b.

Protein separation was first examined with Bovine Serum Albumin (BSA, 66 kDa) and insulin (6 kDa), which are often used as model proteins in biomedical assays³³⁻³⁵. The results show BSA is more difficult to permeate through the small-pore branch, and hence the small-pore branch achieves a purified product of insulin (Supplementary Fig. 13). There is a recent surge in the evidence linking Diabetes mellitus with the pathophysiology of COVID-19^{36,37}. As a demonstration of this contemporary relevance, severe acute respiratory syndrome coronavirus 2 (SARS-CoV-2) nucleocapsid protein (55 kDa) and insulin were tested for the separation. SARS-CoV-2 nucleocapsid protein shows different permeability in the separation channels (Fig. 6c). We found protein-protein and protein-dextran mixtures achieved a better separation compared to dextran-dextran mixture. Likely this is due to dextran having a molecular weight distribution while each protein has a specific molecular weight. With these demonstrations, we can present OM microfluidics as a new platform relevant for biomedical research and applications.”

References (used in the Response Letter for Reviewer#1)

1. Li, X., Ballerini, D. R. & Shen, W. A perspective on paper-based microfluidics: Current status and future trends. *Biomicrofluidics* **6**, 011301 (2012).
2. Nishat, S., Jafry, A. T., Martinez, A. W. & Awan, F. R. Paper-based microfluidics: Simplified fabrication and assay methods. *Sens. Actuator B-Chem.* **336**, 129681 (2021).

Main test, page 24, Figure 5 (a new figure)

Fig. 5 | Manipulation of internal pore size with crosslinking energy. **a**, Photo of OM microchannel made on cover glass. Variation of branch colour is accomplished by changing energy dosage according to the design as specified in Supplementary Fig. 10: the main channel adopted a high energy dosage while the side branch adopted a low energy dosage. **b**, Microscope photo of the same OM microchannel in (a). **c-d**, The corresponding zoom-in microscope photos, where the main channels (labeled as (1), energy dosage: 600 mJ/cm^2) show a more reddish structural colour compared to the side branches (labeled as (2), energy dosage: 300 mJ/cm^2). **f**, A SEM image for the OM microchannel in (d). Red colour is added to highlight the channel. The original image is shown in Supplementary Fig. 12. **g**, The zoom-in SEM cross-section for the main channel in (f). **h**, The zoom-in SEM cross-section for the side branch in (f). Scale bar, (b), $200 \mu\text{m}$, (c-e), $100 \mu\text{m}$, (f), $50 \mu\text{m}$, (g, h), 200 nm .

Main test, page 25, Figure 6 (another new figure)

Fig. 6 | Biomolecular separations by OM microchannels. Imaging was conducted by an inverted confocal laser scanning microscope (CLSM, Nikon AIR). **a**, Separation of two polysaccharides of different molecular weights, where the blue and red colours represent 3-kDa (Cascade Blue labeled) and 70-kDa (Texas Red labeled) fluorescent dextran, respectively. **b**, Separation of polysaccharide from protein, where the blue and red colours represent a fluorescent dextran (3 kDa) and a red fluorescent protein (RFP, 27 kDa), respectively. **c**, Separation of two proteins of different molecular weight, where the green and red colours represent SARS-CoV-2 nucleocapsid protein (55 kDa, pre-stained with the dye SYPRO Orange) and fluorescent insulin (6 kDa, Alexa Fluor 680 labeled), respectively. These OM microchannels were made on cover glass using a micro-LED instrument ($\lambda_i = 405 \text{ nm}$). Microchannels were printed according to the design in Supplementary Fig. 10 with different energy dosages in the main channel and the side branch. The microscope photos of these OM channels are shown in Fig. 5c-e, respectively. Further experimental details are provided in Supplementary Table 6. Scale bar, (a-c), $100 \mu\text{m}$.

Response (in blue) to comments (in black) from Reviewer #2 (text change highlight in yellow and turquoise)

The current manuscript extends the work presented by the authors in their 2019 Nature paper (Nature, 2019, 570, 363), with periodic porous structures being printed on flexible substrates to create self-enclosed channels for capillary flow. In this current work, both fluid velocity and structural color variations within microfluidic channels are found to be related to the viscosity and refractive index of the solvents used, thus providing a basis for in-situ sensing. Additionally, the current studies report the use of simpler, cheaper and higher resolution fabrication methods for fabricating capillary microfluidic chips. The work is generally well-presented and the results nice. That said, the sensing aspects of the work appear most interesting to me, but at the moment only simple proof-of-principle studies are shown. This is disappointing and it would be far more useful for the authors to show sensing operation using real world samples. Accordingly, and since the underlying concepts of the technical approach have been published previously, I would recommend that the authors provide a revised manuscript that provides much more relevant sensing data. This should address real-world samples and benchmark performance with respect to existing sensing methods.

We appreciate that the reviewer acknowledged our meaningful contribution to the development of a simpler, cheaper and high-resolution technology for making capillary microfluidic chips. We also thank the reviewer's comments that have driven us to expand the scope of our study.

As recommended by the reviewer, we focused on utilizing OM technology for a number of "real world samples". We set out to realize a big enabling leap for our technology: making the OM channels compatible with water-based flow. We achieved this by identifying water-based solutions that are compatible with our polymer structures. This enables the extension of our new technology to biomedically relevant research. Specifically, we demonstrated pore-size based separations for polysaccharides (dextrans) and several proteins (Figs. 5 and 6 and Supp. Fig. 12). Regarding protein samples, we first tested red fluorescent protein, insulin, and Bovine Serum Albumin (BSA). Moreover, as a demonstration of contemporary relevance, we showed the separation of insulin from SARS-CoV-2 nucleocapsid protein in OM microchannel. It is worth noting that there has been a recent surge in evidence linking the role of Diabetes mellitus to the pathophysiology of COVID-19^{3,4}. New figures (Figs. 5 and 6) have been appended to the end of this response letter.

Meanwhile, we have added more sensing data based on the reviewer's suggestions. In the revised manuscript, we studied viscosity sensing with two classes of solvents (alkanes and alcohols), both with pure solvents and mixtures (revised Fig. 3e). We added a plot showing that Bragg peak intensity could be used as the indicator for refractive index sensing, apart from the peak position (revised Fig. 4d). In addition, we demonstrated the ability of OM microfluidics to detect solute concentration in aqueous solution based on refractive index sensing (Supp. Fig. 8). We would like to present the details in the following point-by-point responses.

Specific Comments

1. The authors are successful in fabricating structures on flexible substrates, but do not explain key features or benefits. For example, in Fig. 1c and Fig. 4c, it seems likely that the bending of the substrate influences the spectra (the color is not uniform across the whole substrate).

We would like to make a clarification: the non-uniform colour is

a, b, Photos for an OM microfluidics on PET sheet, in bent and flat status, respectively. **c, d**, Photos for an OM film, sticking on finger and floating on water, respectively. Scale bar, **a,b**, 1 cm; **c,d**, 0.5 cm.

from the OM samples themselves, not from the bending, as shown by the photos above. Moreover, in the revised manuscript, we studied the impact of bending on the OM reflectance spectrum. The result shows that OM microfluidics is able to maintain the spectral characteristics with negligible change upon the bending (main text page 8, line 169-171; Supp. Fig. 9).

2. The authors present a method for sensing the viscosity of a liquid by monitoring the velocity of flow in the channels (Fig. 3e). The tests are carried out using a pure hydrocarbon. How will the measurement be effected if the sample is a combination of multiple chemicals? Is the capillary flow velocity only dependent on the viscosity? Will it still show the same trend with other types of solvents?

We appreciate the reviewer's comment and have expanded our discussion of viscosity sensing. In the revised Fig. 3e, we conducted more experiments on viscosity sensing and added three datasets. Firstly, we studied the relationship between the microfluidic flow parameter (dL^2/dt) and the viscosity of alkane mixtures. Secondly, we extended the study to pure alcohols, another class of solvents. Thirdly, we studied the relationship between dL^2/dt and alcohol mixtures. Together with the original data of pure alkanes, all four datasets demonstrate that the OM flow parameter (dL^2/dt) is inversely proportional to liquid viscosity. Moreover, OM microfluidics is able to make the distinction between the two solvent classes based on viscosity sensing (main text, page 7, 148-154).

We found OM flow behavior follows the Lucas-Washburn model (see the equation in main text page 6). In detail, the flow parameter (dL^2/dt) is related to OM internal pore structure, liquid - OM channel surface interaction, and liquid viscosity. We had provided the formula derivation in Supplementary Discussion. In Fig. 3d, we made OM channels of different internal pore sizes (as indicated by Bragg peak position), for studying the effect of pore structure on flow parameter (liquid was kept as *n*-hexadecane). In the revised Fig. 3e, we showed OM flow parameter (dL^2/dt) is inversely proportional to liquid viscosity (OM channel conditions were fixed: polystyrene/9,10-phenanthrenequinone, Bragg peak = 670 nm).

3. Similarly, in the refractive index testing, the authors choose alcohol to prove the relationship. What happens when the solvent is changed?

We would like to thank the reviewer for suggesting the extension of refractive index sensing to other solvents. In the revised manuscript, we studied refractive index sensing with water as a new class of solvent and acrylamide as the solute. We demonstrated the ability of OM microfluidics to detect solute concentrations based on liquid refractive index (main text page 8, line 165 – 166; Supp. Fig. 8).

4. The authors characterize the refractive index via the Bragg peak (Fig. 4b). But the peak intensity varies more than the Bragg peak. Why would one not choose the peak intensity as the indicator?

We appreciate the reviewer's comment and we have analyzed the relationship between peak intensity and liquid refractive index. We conducted the analysis and added the plots in revised Fig. 4d as well as Supp. Fig. 8b. In the revised main text, we pointed out that both the position and the intensity of the Bragg peak can be used as the indicator for refractive index sensing (main text page 8, line 163-164).

References (used in the Response Letter for Reviewer#2)

3 Muller, J. A. et al. SARS-CoV-2 infects and replicates in cells of the human endocrine and exocrine pancreas. *Nat.*

Main test, page 24, Figure 5 (a new figure)

Fig. 5 | Manipulation of internal pore size with crosslinking energy. **a**, Photo of OM microchannel made on cover glass. Variation of branch colour is accomplished by changing energy dosage according to the design as specified in Supplementary Fig. 10: the main channel adopted a high energy dosage while the side branch adopted a low energy dosage. **b**, Microscope photo of the same OM microchannel in (a). **c-d**, The corresponding zoom-in microscope photos, where the main channels (labeled as (1), energy dosage: 600 mJ/cm²) show a more reddish structural colour compared to the side branches (labeled as (2), energy dosage: 300 mJ/cm²). **f**, A SEM image for the OM microchannel in (d). Red colour is added to highlight the channel. The original image is shown in Supplementary Fig. 12. **g**, The zoom-in SEM cross-section for the main channel in (f). **h**, The zoom-in SEM cross-section for the side branch in (f). Scale bar, (b), 200 μ m, (c-e), 100 μ m, (f), 50 μ m, (g, h), 200 nm.

Main test, page 25, Figure 6 (another new figure)

Fig. 6 | Biomolecular separations by OM microchannels. Imaging was conducted by an inverted confocal laser scanning microscope (CLSM, Nikon A1R). **a**, Separation of two polysaccharides of different molecular weights, where the blue and red colours represent 3-kDa (Cascade Blue labeled) and 70-kDa (Texas Red labeled) fluorescent dextran, respectively. **b**, Separation of polysaccharide from protein, where the blue and red colours represent a fluorescent dextran (3 kDa) and a red fluorescent protein (RFP, 27 kDa), respectively. **c**, Separation of two proteins of different molecular weight, where the green and red colours represent SARS-CoV-2 nucleocapsid protein (55 kDa, pre-stained with the dye SYPRO Orange) and fluorescent insulin (6 kDa, Alexa Fluor 680 labeled), respectively. These OM microchannels were made on cover glass using a micro-LED instrument ($\lambda_i = 405$ nm). Microchannels were printed according to the design in Supplementary Fig. 10 with different energy dosages in the main channel and the side branch. The microscope photos of these OM channels are shown in Fig. 5c-e, respectively. Further experimental details are provided in Supplementary Table 6. Scale bar, (a-c), 100 μ m.

REVIEWER COMMENTS

Reviewer #1 (Remarks to the Author):

I'd like to commend the authors for the two demonstrations (Fig. 5 & 6) added in the revisions in response to reviewers' comments which now demonstrate that OM can be superior to paper for microfluidics. Impressive last-mile effort! I recommend acceptance.

Albert Folch (Reviewer #1)

Reviewer #3 (Remarks to the Author):

After revision, the authors did quite a few works to improve the quality of paper. However, there are still some questions confused me.

- 1、 As the Reviewer #1 claimed the OM microfluidics is a very expensive way for making capillary flows on plastic. The authors particularly elaborated an advantage of OM microfluidics over paper-based platforms. The advantages in practicability are still not clear.
- 2、 The author has proved that substrate bending has little effect on spectral characteristics. In my opinion, stress will inevitably lead to the change of OM structure, and the change of effective refractive index will definitely affect the spectral characteristics. Can the authors give a detailed explanation?

Response (in blue) to comments (in black) from Reviewer #3 (text change is highlighted in yellow)

After revision, the authors did quite a few works to improve the quality of paper. However, there are still some questions confused me.

We greatly appreciate that the reviewer acknowledged our previous effort to improve the quality of this manuscript. In the past weeks, we focused on further revising our manuscript with a series of systematic experiments to address all the reviewer's concerns.

1. As the Reviewer #1 claimed the OM microfluidics is a very expensive way for making capillary flows on plastic. The authors particularly elaborated an advantage of OM microfluidics over paper-based platforms. The advantages in practicability are still not clear.

We thank the reviewer for the feedback on this point. While we acknowledge that the practicability could vary depending on the particular applications, we would like to bring to the kind notice of the reviewer that OM microfluidics has certain practical advantages due to its ability to be used in different conditions. Considering that this is a new approach to making microfluidic devices, we present some scenarios where OM microfluidics may have more practical advantages:

- (1) The OM process fast-tracks the photolithography methods, obviating some of the steps required in traditional microfluidics fabrication. This makes it practical for faster prototyping and testing of different designs. We made the relevant change in the revised main text, which is highlighted below.
- (2) As described in the main text, OM microfluidics is amenable to high-resolution position control of pore size, which is challenging to accomplish in conventional paper microfluidics. We have described this in the main text (line 14–17, 36–38, 243–244).
- (3) Another merit to consider is that the OM method allows easy fabrication of nanoporous channels in polymers (e.g. polystyrene, polycarbonate) that is challenging to accomplish otherwise. We have reflected this in the main text (line 239–241).

In addition, the below table comparing OM microfluidics and paper microfluidics could provide some further insights.

Table 1. The comparison between OM microfluidics and paper microfluidics.

Property	OM Microfluidics	Paper Microfluidics
Transparency	Transparent/ Translucent	Opaque
Inverted microscope	Compatible	Not compatible
Film Thickness	1–2 μm	$\sim 100 \mu\text{m}^1$
Channel Width	$\geq 5 \mu\text{m}$	$> 60 \mu\text{m}^{2,3}$
Pore Size	10–200 nm (narrowly distributed)	2–25 μm^+ (widely distributed) ⁴
Pore size control	Possible	Not possible / difficult
Materials	Polystyrene, etc.	Cellulose

References:

- 1 Mahmud, M. A., Blondeel, E. J. M., Kaddoura, M. & MacDonald, B. D. Features in Microfluidic Paper-Based Devices Made by Laser Cutting: How Small Can They Be? *Micromachines* **9** (2018).

- 2 Li, X., Ballerini, D. R. & Shen, W. A perspective on paper-based microfluidics: Current status and future trends. *Biomicrofluidics* **6**, 011301 (2012).
- 3 Nishat, S., Jafry, A. T., Martinez, A. W. & Awan, F. R. Paper-based microfluidics: Simplified fabrication and assay methods. *Sens. Actuator B-Chem.* **336**, 129681 (2021).
- 4 Fernandes, S. C., Walz, J. A., Wilson, D. J., Brooks, J. C. & Mace, C. R. Beyond Wicking: Expanding the Role of Patterned Paper as the Foundation for an Analytical Platform. *Anal. Chem.* **89**, 5655-5665 (2017).

[†]Based on Whatman filter paper pore sizes.

In the revised main text, we reflected the changes regarding this comment in the Discussion section, which are highlighted in below:

Main text, page 11–12, line 237–248,

“Generating an advanced platform technology for the creation of microfluidic devices represents a game-changing opportunity for the future of wearable, analytical and sensing technologies. We have demonstrated the creation of microfluidic devices using a novel photolithography technique generally applicable to commodity polymers such as polystyrene, poly(methyl methacrylate), as well as polycarbonate. We have also demonstrated the capillary flow of several classes of liquids including aqueous solutions in OM microfluidic channels. The ease of combining differing internal porosity in a single miniature OM device enables pore-size sieving functions and applications. This approach removes some of the steps typically required in conventional microfluidic devices, allowing quick development of microfluidic designs on multiple substrate types. Our microfluidic devices provide an opportunity to develop applications for filtration, skin contact technology, and in situ sensing and analysis.”

2. The author has proved that substrate bending has little effect on spectral characteristics. In my opinion, stress will inevitably lead to the change of OM structure, and the change of effective refractive index will definitely affect the spectral characteristics. Can the authors give a detailed explanation?

We thank the reviewer’s comment on the bending of OM microfluidics. In general, we agree with the reviewer that the change in the OM structure will change its optical characteristics. The intended point in the manuscript was that the bending did not damage or permanently deform the structure; thus, the spectrum before and after the bending remained unchanged. We now see the phrasing of this point was unclear.

We have since done additional work to clarify the impact of bending. We conducted in situ spectral and optical measurements of the OM films under bending conditions. Based on these experimental results, we have revised our statement on bending (main text: line 169–175) and presented the detailed explanation in the revised Suppl. Fig. 9. The revisions are summarized as follows:

We have shown that the OM microfluidic films made by polycarbonate on silicon wafer can be peeled off from the substrate (Fig. 4e). The fabricated OM films via the previously reported method were subsequently transferred onto the aluminium foil (60- μm -thick) for the bending tests. Aluminium foil was chosen for its high reflectance and compatibility as the reference material in our spectrometer. A vice was used to bend the OM film/aluminium samples, and profile photos were

taken to measure sample curvature. The in situ reflectance spectrum was measured as the bending of OM film was increased. As an alternative approach to check the impact of bending, DSLR photos were taken for colour comparison under different bending conditions.

In the measured samples, the spectrum (in particular the peak position) did not change significantly. In situ measurements show negligible spectral change when the curvature of OM film is $\leq 0.1 \text{ mm}^{-1}$. At high curvatures *i.e.* up to 0.5 mm^{-1} , the Bragg peak shift of OM films remains in the small range of 1–4 nm (revised Suppl. Fig. 9a, b). These results are consistent with the observation from the DSLR photos, which indicate the OM film kept $< 3 \text{ nm}$ colour shift as the curvature increases to 0.5 mm^{-1} (revised Suppl. Fig. 9c–g). In addition, after releasing the bending and placing the OM film back onto silicon wafer, the reflectance spectrum remains almost unchanged in comparison to its pristine status (revised Suppl. Fig. 9h). Thus we conclude that our bending conditions did not permanently deform the OM structure under this condition.

Eventually there will be a point where the stress/strain is enough to deform the structure and change the spectrum permanently and this depends on a number of factors including the molecular weight of the polymer. The maximum curvature tested was 0.5 mm^{-1} (radius of curvature: 2 mm). With the OM film thickness being 1–2 μm and the aluminium foil thickness being 60 μm , the maximum strain applied is estimated as 1.5% (strain = distance from neutral axis / radius of curvature). We expect that more extreme bending is required to achieve the required strain for deforming the OM structure. The study of such extreme bending conditions are not the major purpose of this paper, thus we would like to report those studies in a separate article in future.

In the revised manuscript, we clearly specify the experimental conditions on our bending statements. The revisions made in the main text and Suppl. Info. provide further insights, which are appended as follows:

Main text, page 8, line 169–175, the description about mechanical bending experiments is revised as follows:

“OM films made by polycarbonate were transferred onto the aluminium foil (60- μm thickness) to study the effect of bending on the structural colour. The in situ spectrum shows negligible colour change when the curvature of the film is $\leq 0.1 \text{ mm}^{-1}$. As the curvature further increases to 0.5 mm^{-1} , the Bragg peak shift of OM film remains in the small range of 1–4 nm. The spectrum remains almost unchanged after releasing the bending and returning the film to silicon wafer (see Supplementary Fig. 9 for details).”

Suppl. Info., page 10–11, a revised Supplementary Fig. 9 is presented (please see the next page):

Supplementary Fig. 9 | Impact of bending on the structural colour of OM film. **a**, In situ reflectance spectrum of the OM film during the bending test. The spectrum reference was aluminium (LUXAL UV, 60- μ m-thick, Toyo Aluminium K.K.). **b**, Bragg peak shift versus the curvatures of OM film. **c, d**, Side-view and top-view photos of an OM film on the aluminium foil before the bending test. **e, f**, Side-view and top-view photos of the film during the bending test. **g**, Hue and colour shift versus the curvatures of OM film. **h**, In the end, bending was released and the OM film was placed back onto silicon wafer for spectral comparison. Scale bar in (c) and (e), 10 mm. The OM films were made using Polycarbonate (PC)/4,4'-bis-(diethylamino)-benzophenone

(BDABP) on silicon wafer (LED $\lambda_i = 385$ nm). The film was peeled off from the substrate carefully in water and transferred onto a piece of aluminium foil. A vice was used for the bending to apply various curvatures to the OM film. As the bending was increased, the in situ reflectance spectrum was recorded. Side-view photos were taken at the same time to obtain the curvature. The curvature was calculated by the software *Fiji* (version 2.1.0/1.53c) using the built-in plugin *Kappa*. Reported values are the average curvature of a 2.5-mm-length curve at the apex region of the bent film where its reflectance spectrum was recorded. In situ measurements show little spectral change when the curvature of OM film is ≤ 0.1 mm⁻¹ (radius of curvature: ≥ 10 mm). As the curvature further increases to 0.5 mm⁻¹ (radius of curvature decreases to 2 mm), the Bragg peak shift of OM film remains in the small range of 1–4 nm.

As an alternative approach to check the impact of bending, top-view photos were taken for colour comparison under different bending conditions. To keep consistency, an 18% neutral grey card was used as the colour reference. Colour comparison focused on the apex region of the bent OM film that held a fixed angle to the camera lens. The wavelength value of the colour is derived from the hue value according to the visible light spectrum chart, using *ImageJ* software (version 1.52p). Photo comparison indicates that the colour shift of OM film keeps < 3 nm as its curvature increases to 0.5 mm⁻¹. This is consistent with the results from the spectral measurements.

The results show negligible change of structural colour in this range of bending conditions. In addition, after releasing the bending and placing the OM film back onto silicon wafer, the reflectance spectrum remains almost unchanged in comparison to its pristine status. This indicates bending to such an extent does not permanently deform the OM structure.

REVIEWERS' COMMENTS

Reviewer #3 (Remarks to the Author):

After revision, the authors did quite a few works to improve the quality of paper. I would recommend this manuscript for this format to be accepted and published.

REVIEWER COMMENTS

Reviewer #1 (Remarks to the Author):

This paper reports the application of the OM technique pioneered by Sivaniah's group to microfluidics. It makes sense -- since OM generates a porous layer using photolithography, it follows that it should be possible to generate porosities that are useful for capillary flow.

I enjoyed the cleverness of the paper, the paper is well written, and I think it would make a great Lab-on-a-chip-style paper, but for Nature Communications, I was expecting something that went beyond the demonstration of flow in the OM-patterned structures. It feels like we are reading about a very expensive way of making capillary flows on plastic, when one could use paper instead. The authors need to think about a "killer app" for which OM is optimally suited and generate one more figure with it in order to be suitable for publication in Nature Comm.

Reviewer #2 (Remarks to the Author):

The current manuscript extends the work presented by the authors in their 2019 Nature paper (Nature, 2019, 570, 363), with periodic porous structures being printed on flexible substrates to create self-enclosed channels for capillary flow. In this current work, both fluid velocity and structural color variations within microfluidic channels are found to be related to the viscosity and refractive index of the solvents used, thus providing a basis for in-situ sensing. Additionally, the current studies report the use of simpler, cheaper and higher resolution fabrication methods for fabricating capillary microfluidic chips. The work is generally well-presented and the results nice. That said, the sensing aspects of the work appear most interesting to me, but at the moment only simple proof-of-principle studies are shown. This is disappointing and it would be far more useful for the authors to show sensing operation using real world samples. Accordingly, and since the underlying concepts of the technical approach have been published previously, I would recommend that the authors provide a revised manuscript that provides much more relevant sensing data. This should address real-world samples and benchmark performance with respect to existing sensing methods.

Specific Comments

1.

The authors are successful in fabricating structures on flexible substrates, but do not explain key features or benefits. For example, in Fig. 1c and Fig. 4c, it seems likely that the bending of the substrate influences the spectra (the color is not uniform across the whole substrate).

2.

The authors present a method for sensing the viscosity of a liquid by monitoring the velocity of flow in the channels (Fig. 3e). The tests are carried out using a pure hydrocarbon. How will the measurement be effected if the sample is a combination of multiple chemicals? Is the capillary flow velocity only dependent on the viscosity? Will it still show the same trend with other types of solvents?

3.

Similarly, in the refractive index testing, the authors choose alcohol to prove the relationship. What happens when the solvent is changed?

4.

The authors characterize the refractive index via the Bragg peak (Fig. 4b). But the peak intensity varies more than the Bragg peak. Why would one not choose the peak intensity as the indicator?

Point-by-Point Response

Overall response

We greatly appreciate both reviewers' complimentary and constructive comments that helped improve the quality of our manuscript. As head of a research group that emphasizes practical implementation of our research findings through entrepreneurship, I appreciate that the comments highlighted the practical aspects of our work. We took these comments as a renewed call-to-action.

Over the past six months, we have focused on addressing the crucial points emphasized by the reviewers (Reviewer #1: "killer app"; Reviewer #2: "real world samples" and additional results on sensing phenomenon).

We are pleased to announce that we have achieved two major breakthroughs through substantial effort:

- (1) Submicron-scale manipulation and combination of different porosities in a single miniature device, which enables pore-size based applications.
- (2) In-channel water flow, which open new vistas of opportunities for OM microfluidics in biomedically relevant research and applications, such as the separation of various biomolecules (*e.g.*, polysaccharides, proteins including BSA, insulin, SARS-CoV-2 nucleocapsid protein).

I would like to emphasize the second breakthrough. We also felt that our first submission lacked the "killer-app" status due to lack of biologically relevant demonstrations. Microfluidics will have a broad range of application, including the oil-and-gas and fine-chemical synthesis sectors. However, the dominant application sector is in health sensing, medical diagnosis and wearables. Despite the trial of various surfactants in the preceding 4 years, we had not succeeded to achieve sufficient capillary flow of waterborne biomolecules and biomarkers into our OM microchannels.

We are grateful for the reviewers' feedback that led us to make major experimental additions to the paper in the intervening 6 months. We began a collaboration with a biomedical group who suggested a number of commonly used materials in biological assays. These additions suitably enhanced the hydrophilicity of OM microchannels. They were also able to recommend a number of real-world biomolecules that might have interest to the microfluidic biological assay community.

In the revised version of our manuscript, we included demonstrations of these two breakthroughs along with two new figures (Figs. 5 and 6).

In addition, we made another noteworthy improvement: the printing of OM microchannels on transparent substrates *e.g.* cover glass (Our previous publication *Nature* 363, 579 introduced micro-printing on silicon wafer only). This achievement facilitated the observation of OM microfluidic devices by inverted microscopes, which are commonly used for microfluidics flow tests, as shown in the new figures.

With these new findings and other impactful results on structural colour-enabled sensing capability, we are confident that we managed to show the versatility of the OM microfluidics platform, potentially opening up new horizons for the concept of microfluidics.

Please find our replies for Reviewer #1 and Reviewer #2 in a point-by-point format below. For enhanced clarity, we colour-coded the text change highlights in (i) yellow for Figures 5 and 6 (in response to both Reviewers), and (ii) turquoise for responding to additional comments of Reviewer #2, and (iii) light gray for the other common changes and general text improvements.

Best regards,
Prof. Easan Sivaniah

Response (in blue) to comments (in black) from Reviewer #1 (text change highlight in yellow)

This paper reports the application of the OM technique pioneered by Sivaniah's group to microfluidics. It makes sense -- since OM generates a porous layer using photolithography, it follows that it should be possible to generate porosities that are useful for capillary flow.

I enjoyed the cleverness of the paper, the paper is well written, and I think it would make a great Lab-on-a-chip-style paper, but for Nature Communications, I was expecting something that went beyond the demonstration of flow in the OM-patterned structures. It feels like we are reading about a very expensive way of making capillary flows on plastic, when one could use paper instead. The authors need to think about a "killer app" for which OM is optimally suited and generate one more figure with it in order to be suitable for publication in Nature Comm.

We appreciate the reviewer's comment on finding an optimally suited "killer app" of OM technology, which motivated us to further explore this aspect of our technology for most of this year.

After making a series of breakthroughs, we are now able to show that OM microfluidics is able to generate differing internal porosities at submicron scale in a single miniature device (Fig. 5). By making use of this functionality, we could also demonstrate pore-size based separation of biomolecules (Fig. 6).

We think that these two achievements demonstrate an advantage of OM microfluidics over paper-based platforms. The convenience of pore size variation at submicron scale in a well-defined way is a unique feature of OM microfluidic technology, which cannot be achieved by paper microfluidics due to their limitations in device resolution and internal pore size control^{1,2}.

Plus, combined with the original finding of structural colour enabled sensing capabilities, now we can present OM microfluidics as a new and versatile platform relevant for a wide range of research topics and hopefully practical applications.

Overall, we hope that the reviewer will acknowledge the fact that the technology we present is not limited to a single "killer app" but brings a new perspective to device design for microfluidics. The revisions made in the main text provide further insights, which are as follows:

Main text, page 8-11, line 172-231:

“Porosity manipulation and biomolecular separations. Up to this point, the microfluidic flow has been demonstrated mainly with alcohols and alkanes, due to their affinity to polystyrene. However, water is the most commonly used liquid for microfluidics as it is a solvent for numerous biomolecules. Achieving aqueous flow in OM microfluidics would extend its scope to biomedical applications. The challenge is that the polystyrene OM is not hydrophilic (water contact angle greater than 90°, Supplementary Fig. 3a). To address this problem, we identified an additive that can improve the affinity between the polystyrene OM channels and water-based systems. Acrylamide, a commodity chemical frequently used in biomedical research and industry, including as a material for electrophoresis gels, was found to be effective in enabling aqueous flow in OM channels. The water/acrylamide system has a contact angle (time = 0 s) of 63° on polystyrene OM surface, reducing to 30° within one minute (Supplementary Fig. 3b). The improved affinity allows for aqueous flow through the OM channels (Fig. 2d, flow marker is a green fluorescent protein). This capability enables OM microfluidics to be used in biomedically relevant tests which use an aqueous environment.

The advantage of OM technology for biomedical applications is the convenience for making varying porosity at submicron scale in a single miniature device, while the differing porosities are correlated to its

structural colours and can be tailored for specific purposes. Through controlling the crosslinking energy at the local regions of the print pattern, the internal porosity of the OM microchannels can be varied and combined. As a demonstration of this concept, one can supply a high energy dose to the main channel and lower doses to the side branches (further details in Supplementary Fig. 10). With this design, OM microchannels with pore size variation were printed on cover glass (Fig. 5a, b, Supplementary Fig. 11). The microscope photos in Figs. 5 c-e clearly show that the main channels (high energy dosage) have a more reddish structural colour than the side branches (low energy dosage). Moreover, SEM cross-sections show that the channel with high energy dosage contains larger internal pores (Fig. 5f-h, Supplementary Fig. 12), confirming the positive correlation between structural colour and internal pore size. In addition, in Fig. 5 the microchannels of pore-varying features were printed on glass substrate using the micro-LED instrument. Note that the channels were printed on glass to facilitate viewing by inverted microscopes.

The capillary flow of biomolecules in such microchannels was performed and proof-of-concept studies were conducted to demonstrate the pore-size based separation functionality. Polysaccharides and proteins of various molecular weights were selected to prepare the aqueous solutions of mixing solutes. The aqueous solution was introduced to the open cross-section of the OM main channel and could flow into the separation region with branch of smaller pore size. The visualization of biomolecules separation was conducted by an inverted CLSM, and the biomolecule solutes can be differentiated by their fluorescence (see details in the Methods section and Supplementary Table 6). Fig. 6a shows the separation of 3-kDa dextran from 70-kDa dextran in a Y-type channel with different porosity in each branch. In both branches, 3-kDa dextran permeates faster than the 70-kDa one. 70-kDa dextran permeates faster in the large-pore branch compared to the small-pore branch. This indicates 70-kDa dextran encounters more resistance in the small-pore branch. These results demonstrate that the OM microfluidic channels can distinguish biomolecules by controlling the internal pore sizes, and this can be achieved in submillimeter length scale with a micrometer-scale printing resolution. Stronger separation ability can be observed for the separation of a low molecular-weight dextran (3 kDa) from a large molecular-weight protein (red fluorescent protein, 27 kDa), see Fig. 6b.

Protein separation was first examined with Bovine Serum Albumin (BSA, 66 kDa) and insulin (6 kDa), which are often used as model proteins in biomedical assays³³⁻³⁵. The results show BSA is more difficult to permeate through the small-pore branch, and hence the small-pore branch achieves a purified product of insulin (Supplementary Fig. 13). There is a recent surge in the evidences linking Diabetes mellitus with the pathophysiology of COVID-19^{36,37}. As a demonstration of this contemporary relevance, Severe acute respiratory syndrome coronavirus 2 (SARS-CoV-2) nucleocapsid protein (55 kDa) and insulin were tested for the separation. SARS-CoV-2 nucleocapsid protein shows different permeability in the separation channels (Fig. 6c). We found protein-protein and protein-dextran mixtures achieved a better separation compared to dextran-dextran mixture. Likely this is due to dextran having a molecular weight distribution while each protein has a specific molecular weight. With these demonstrations, we can present OM microfluidics as a new platform relevant for biomedical research and applications.”

References (used in the Response Letter for Reviewer#1)

1. Li, X., Ballerini, D. R. & Shen, W. A perspective on paper-based microfluidics: Current status and future trends. *Biomicrofluidics* **6**, 011301 (2012).
2. Nishat, S., Jafry, A. T., Martinez, A. W. & Awan, F. R. Paper-based microfluidics: Simplified fabrication and assay methods. *Sens. Actuator B-Chem.* **336**, 129681 (2021).

Main test, page 24, Figure 5 (a new figure)

Fig. 5 | Manipulation of internal pore size with crosslinking energy. **a**, Photo of OM microchannel made on cover glass. Variation of branch colour is accomplished by changing energy dosage according to the design as specified in Supplementary Fig. 10: the main channel adopted a high energy dosage while the side branch adopted a low energy dosage. **b**, Microscope photo of the same OM microchannel in (a). **c-d**, The corresponding zoom-in microscope photos, where the main channels (labeled as (1), energy dosage: 600 mJ/cm^2) show a more reddish structural colour compared to the side branches (labeled as (2), energy dosage: 300 mJ/cm^2). **f**, A SEM image for the OM microchannel in (d). Red colour is added to highlight the channel. The original image is shown in Supplementary Fig. 12. **g**, The zoom-in SEM cross-section for the main channel in (f). **h**, The zoom-in SEM cross-section for the side branch in (f). Scale bar, (b), $200 \mu\text{m}$, (c-e), $100 \mu\text{m}$. (f), $50 \mu\text{m}$. (g, h), 200 nm .

Main test, page 25, Figure 6 (another new figure)

Fig. 6 | Biomolecular separations by OM microchannels. Imaging was conducted by an inverted confocal laser scanning microscope (CLSM, Nikon AIR). **a**, Separation of two polysaccharides of different molecular weights, where the blue and red colours represent 3-kDa (Cascade Blue labeled) and 70-kDa (Texas Red labeled) fluorescent dextran, respectively. **b**, Separation of polysaccharide from protein, where the blue and red colours represent a fluorescent dextran (3 kDa) and a red fluorescent protein (RFP, 27 kDa), respectively. **c**, Separation of two proteins of different molecular weight, where the green and red colours represent SARS-CoV-2 nucleocapsid protein (55 kDa, pre-stained with the dye SYPRO Orange) and fluorescent insulin (Alexa Fluor 680 labeled, 6 kDa), respectively. These OM microchannels were made on cover glass using a micro-LED instrument ($\lambda_i = 405 \text{ nm}$). Microchannels were printed according to the design in Supplementary Fig. 10 with different energy dosages in the main channel and the side branch. The microscope photos of these OM channels are shown in Fig. 5c-e, respectively. Further experimental details are provided in Supplementary Table 6. Scale bar, (a-c), $100 \mu\text{m}$.

Response (in blue) to comments (in black) from Reviewer #2 (text change highlight in yellow and turquoise)

The current manuscript extends the work presented by the authors in their 2019 Nature paper (Nature, 2019, 570, 363), with periodic porous structures being printed on flexible substrates to create self-enclosed channels for capillary flow. In this current work, both fluid velocity and structural color variations within microfluidic channels are found to be related to the viscosity and refractive index of the solvents used, thus providing a basis for in-situ sensing. Additionally, the current studies report the use of simpler, cheaper and higher resolution fabrication methods for fabricating capillary microfluidic chips. The work is generally well-presented and the results nice. That said, the sensing aspects of the work appear most interesting to me, but at the moment only simple proof-of-principle studies are shown. This is disappointing and it would be far more useful for the authors to show sensing operation using real world samples. Accordingly, and since the underlying concepts of the technical approach have been published previously, I would recommend that the authors provide a revised manuscript that provides much more relevant sensing data. This should address real-world samples and benchmark performance with respect to existing sensing methods.

We appreciate that the reviewer acknowledged our meaningful contribution to the development of a simpler, cheaper and high-resolution technology for making capillary microfluidic chips. We also thank the reviewer's comments that have driven us to expand the scope of our study.

As recommended by the reviewer, we focused on utilizing OM technology for a number of "real world samples". We set out to realize a big enabling leap for our technology: making the OM channels compatible with water-based flow. We achieved this by identifying water-based solutions that are compatible with our polymer structures. This enables the extension of our new technology to biomedically relevant research. Specifically, we demonstrated pore-size based separations for polysaccharides (dextrans) and several proteins (Figs. 5 and 6 and Supp. Fig. 12). Regarding protein samples, we first tested red fluorescent protein, insulin, and Bovine Serum Albumin (BSA). Moreover, as a demonstration of contemporary relevance, we showed the separation of insulin from SARS-CoV-2 nucleocapsid protein in OM microchannel. It is worth noting that there has been a recent surge in evidence linking the role of Diabetes mellitus to the pathophysiology of COVID-19^{3,4}. New figures (Figs. 5 and 6) have been appended to the end of this response letter.

Meanwhile, we have added more sensing data based on the reviewer's suggestions. In the revised manuscript, we studied viscosity sensing with two classes of solvents (alkanes and alcohols), both with pure solvents and mixtures (revised Fig. 3e). We added a plot showing that Bragg peak intensity could be used as the indicator for refractive index sensing, apart from the peak position (revised Fig. 4d). In addition, we demonstrated the ability of OM microfluidics to detect solute concentration in aqueous solution based on refractive index sensing (Supp. Fig. 8). We would like to present the details in the following point-by-point responses.

Specific Comments

1. The authors are successful in fabricating structures on flexible substrates, but do not explain key features or benefits. For example, in Fig. 1c and Fig. 4c, it seems likely that the bending of the substrate influences the spectra (the color is not uniform across the whole substrate).

We would like to make a clarification: the non-uniform colour is

a, b, Photos for an OM microfluidics on PET sheet, in bent and flat status, respectively. **c, d**, Photos for an OM film, sticking on finger and floating on water, respectively. Scale bar, **a,b**, 1 cm; **c,d**, 0.5 cm.

from the OM samples themselves, not from the bending, as shown by the photos above. Moreover, in the revised manuscript, we studied the impact of bending on the OM reflectance spectrum. The result shows that OM microfluidics is able to maintain the spectral characteristics with negligible change upon the bending (main text page 8, line 169-171; Supp. Fig. 9).

2. The authors present a method for sensing the viscosity of a liquid by monitoring the velocity of flow in the channels (Fig. 3e). The tests are carried out using a pure hydrocarbon. How will the measurement be effected if the sample is a combination of multiple chemicals? Is the capillary flow velocity only dependent on the viscosity? Will it still show the same trend with other types of solvents?

We appreciate the reviewer's comment and have expanded our discussion of viscosity sensing. In the revised Fig. 3e, we conducted more experiments on viscosity sensing and added three datasets. Firstly, we studied the relationship between the microfluidic flow parameter (dL^2/dt) and the viscosity of alkane mixtures. Secondly, we extended the study to pure alcohols, another class of solvents. Thirdly, we studied the relationship between dL^2/dt and alcohol mixtures. Together with the original data of pure alkanes, all four datasets demonstrate that the OM flow parameter (dL^2/dt) is inversely proportional to liquid viscosity. Moreover, OM microfluidics is able to make the distinction between the two solvent classes based on viscosity sensing (main text, page 7, 148-154).

We found OM flow behavior follows the Lucas-Washburn model (see the equation in main text page 6). In detail, the flow parameter (dL^2/dt) is related to OM internal pore structure, liquid - OM channel surface interaction, and liquid viscosity. We had provided the formula derivation in Supplementary Discussion. In Fig. 3d, we made OM channels of different internal pore sizes (as indicated by Bragg peak position), for studying the effect of pore structure on flow parameter (liquid was kept as *n*-hexadecane). In the revised Fig. 3e, we showed OM flow parameter (dL^2/dt) is inversely proportional to liquid viscosity (OM channel conditions were fixed: polystyrene/9,10-phenanthrenequinone, Bragg peak = 670 nm).

3. Similarly, in the refractive index testing, the authors choose alcohol to prove the relationship. What happens when the solvent is changed?

We would like to thank the reviewer for suggesting the extension of refractive index sensing to other solvents. In the revised manuscript, we studied refractive index sensing with water as a new class of solvent and acrylamide as the solute. We demonstrated the ability of OM microfluidics to detect solute concentrations based on liquid refractive index (main text page 8, line 165 – 166; Supp. Fig. 8).

4. The authors characterize the refractive index via the Bragg peak (Fig. 4b). But the peak intensity varies more than the Bragg peak. Why would one not choose the peak intensity as the indicator?

We appreciate the reviewer's comment and we have analyzed the relationship between peak intensity and liquid refractive index. We conducted the analysis and added the plots in revised Fig. 4d as well as Supp. Fig. 8b. In the revised main text, we pointed out that both the position and the intensity of the Bragg peak can be used as the indicator for refractive index sensing (main text page 8, line 163-164).

References (used in the Response Letter for Reviewer#2)

3 Muller, J. A. et al. SARS-CoV-2 infects and replicates in cells of the human endocrine and exocrine pancreas. *Nat.*

Main test, page 24, Figure 5 (a new figure)

Fig. 5 | Manipulation of internal pore size with crosslinking energy. **a**, Photo of OM microchannel made on cover glass. Variation of branch colour is accomplished by changing energy dosage according to the design as specified in Supplementary Fig. 10: the main channel adopted a high energy dosage while the side branch adopted a low energy dosage. **b**, Microscope photo of the same OM microchannel in (a). **c-d**, The corresponding zoom-in microscope photos, where the main channels (labeled as (1), energy dosage: 600 mJ/cm²) show a more reddish structural colour compared to the side branches (labeled as (2), energy dosage: 300 mJ/cm²). **f**, A SEM image for the OM microchannel in (d). Red colour is added to highlight the channel. The original image is shown in Supplementary Fig. 12. **g**, The zoom-in SEM cross-section for the main channel in (f). **h**, The zoom-in SEM cross-section for the side branch in (f). Scale bar, (b), 200 μ m, (c-e), 100 μ m, (f), 50 μ m, (g, h), 200 nm.

Main test, page 25, Figure 6 (another new figure)

Fig. 6 | Biomolecular separations by OM microchannels. Imaging was conducted by an inverted confocal laser scanning microscope (CLSM, Nikon A1R). **a**, Separation of two polysaccharides of different molecular weights, where the blue and red colours represent 3-kDa (Cascade Blue labeled) and 70-kDa (Texas Red labeled) fluorescent dextran, respectively. **b**, Separation of polysaccharide from protein, where the blue and red colours represent a fluorescent dextran (3 kDa) and a red fluorescent protein (RFP, 27 kDa), respectively. **c**, Separation of two proteins of different molecular weight, where the green and red colours represent SARS-CoV-2 nucleocapsid protein (55 kDa, pre-stained with the dye SYPRO Orange) and fluorescent insulin (Alexa Fluor 680 labeled, 6 kDa), respectively. These OM microchannels were made on cover glass using a micro-LED instrument ($\lambda_i = 405$ nm). Microchannels were printed according to the design in Supplementary Fig. 10 with different energy dosages in the main channel and the side branch. The microscope photos of these OM channels are shown in Fig. 5c-e, respectively. Further experimental details are provided in Supplementary Table 6. Scale bar, (a-c), 100 μ m.

REVIEWER COMMENTS

Reviewer #1 (Remarks to the Author):

I'd like to commend the authors for the two demonstrations (Fig. 5 & 6) added in the revisions in response to reviewers' comments which now demonstrate that OM can be superior to paper for microfluidics. Impressive last-mile effort! I recommend acceptance.

Albert Folch (Reviewer #1)

Reviewer #3 (Remarks to the Author):

After revision, the authors did quite a few works to improve the quality of paper. However, there are still some questions confused me.

1、 As the Reviewer #1 claimed the OM microfluidics is a very expensive way for making capillary flows on plastic. The authors particularly elaborated an advantage of OM microfluidics over paper-based platforms. The advantages in practicability are still not clear.

2、 The author has proved that substrate bending has little effect on spectral characteristics. In my opinion, stress will inevitably lead to the change of OM structure, and the change of effective refractive index will definitely affect the spectral characteristics. Can the authors give a detailed explanation?

Point-by-Point Response

Response (in blue) to comments (in black) from Reviewer #3 (text change is highlighted in yellow)

After revision, the authors did quite a few works to improve the quality of paper. However, there are still some questions confused me.

We greatly appreciate that the reviewer acknowledged our previous effort to improve the quality of this manuscript. In the past weeks, we focused on further revising our manuscript with a series of systematic experiments to address all the reviewer's concerns.

1. As the Reviewer #1 claimed the OM microfluidics is a very expensive way for making capillary flows on plastic. The authors particularly elaborated an advantage of OM microfluidics over paper-based platforms. The advantages in practicability are still not clear.

We thank the reviewer for the feedback on this point. While we acknowledge that the practicability could vary depending on the particular applications, we would like to bring to the kind notice of the reviewer that OM microfluidics has certain practical advantages due to its ability to be used in different conditions. Considering that this is a new approach to making microfluidic devices, we present some scenarios where OM microfluidics may have more practical advantages:

- (1) The OM process fast-tracks the photolithography methods, obviating some of the steps required in traditional microfluidics fabrication. This makes it practical for faster prototyping and testing of different designs. We made the relevant change in the revised main text, which is highlighted below.
- (2) As described in the main text, OM microfluidics is amenable to high-resolution position control of pore size, which is challenging to accomplish in conventional paper microfluidics. We have described this in the main text (line 14–17, 36–38, 243–244).
- (3) Another merit to consider is that the OM method allows easy fabrication of nanoporous channels in polymers (*e.g.* polystyrene, polycarbonate) that is challenging to accomplish otherwise. We have reflected this in the main text (line 239–241).

In addition, the below table comparing OM microfluidics and paper microfluidics could provide some further insights.

Table 1. The comparison between OM microfluidics and paper microfluidics.

Property	OM Microfluidics	Paper Microfluidics
Transparency	Transparent/ Translucent	Opaque
Inverted microscope	Compatible	Not compatible
Film Thickness	1–2 μm	$\sim 100 \mu\text{m}^1$
Channel Width	$\geq 5 \mu\text{m}$	$> 60 \mu\text{m}^{2,3}$
Pore Size	10–200 nm (narrowly distributed)	2–25 μm^+ (widely distributed) ⁴
Pore size control	Possible	Not possible / difficult
Materials	Polystyrene, etc.	Cellulose

References:

- 1 Mahmud, M. A., Blondeel, E. J. M., Kaddoura, M. & MacDonald, B. D. Features in Microfluidic Paper-Based Devices Made by Laser Cutting: How Small Can They Be? *Micromachines* **9** (2018).

- 2 Li, X., Ballerini, D. R. & Shen, W. A perspective on paper-based microfluidics: Current status and future trends. *Biomicrofluidics* **6**, 011301 (2012).
- 3 Nishat, S., Jafry, A. T., Martinez, A. W. & Awan, F. R. Paper-based microfluidics: Simplified fabrication and assay methods. *Sens. Actuator B-Chem.* **336**, 129681 (2021).
- 4 Fernandes, S. C., Walz, J. A., Wilson, D. J., Brooks, J. C. & Mace, C. R. Beyond Wicking: Expanding the Role of Patterned Paper as the Foundation for an Analytical Platform. *Anal. Chem.* **89**, 5655-5665 (2017).

[†]Based on Whatman filter paper pore sizes.

In the revised main text, we reflected the changes regarding this comment in the Discussion section, which are highlighted in below:

Main text, page 11–12, line 237–248,

“Generating an advanced platform technology for the creation of microfluidic devices represents a game-changing opportunity for the future of wearable, analytical and sensing technologies. We have demonstrated the creation of microfluidic devices using a novel photolithography technique generally applicable to commodity polymers such as polystyrene, poly(methyl methacrylate), as well as polycarbonate. We have also demonstrated the capillary flow of several classes of liquids including aqueous solutions in OM microfluidic channels. The ease of combining differing internal porosity in a single miniature OM device enables pore-size sieving functions and applications. This approach removes some of the steps typically required in conventional microfluidic devices, allowing quick development of microfluidic designs on multiple substrate types. Our microfluidic devices provide an opportunity to develop applications for filtration, skin contact technology, and in situ sensing and analysis.”

2. The author has proved that substrate bending has little effect on spectral characteristics. In my opinion, stress will inevitably lead to the change of OM structure, and the change of effective refractive index will definitely affect the spectral characteristics. Can the authors give a detailed explanation?

We thank the reviewer’s comment on the bending of OM microfluidics. In general, we agree with the reviewer that the change in the OM structure will change its optical characteristics. The intended point in the manuscript was that the bending did not damage or permanently deform the structure; thus, the spectrum before and after the bending remained unchanged. We now see the phrasing of this point was unclear.

We have since done additional work to clarify the impact of bending. We conducted in situ spectral and optical measurements of the OM films under bending conditions. Based on these experimental results, we have revised our statement on bending (main text: line 169–175) and presented the detailed explanation in the revised Suppl. Fig. 9. The revisions are summarized as follows:

We have shown that the OM microfluidic films made by polycarbonate on silicon wafer can be peeled off from the substrate (Fig. 4e). The fabricated OM films via the previously reported method were subsequently transferred onto the aluminium foil (60- μm -thick) for the bending tests. Aluminium foil was chosen for its high reflectance and compatibility as the reference material in our spectrometer. A vice was used to bend the OM film/aluminium samples, and profile photos were taken to measure sample curvature. The in situ reflectance spectrum was measured as the bending of OM film was increased. As an alternative approach to check the impact of bending, DSLR photos were taken for colour comparison under different bending conditions.

In the measured samples, the spectrum (in particular the peak position) did not change significantly. In situ measurements show negligible spectral change when the curvature of OM film is $\leq 0.1 \text{ mm}^{-1}$. At high

curvatures *i.e.* up to 0.5 mm^{-1} , the Bragg peak shift of OM films remains in the small range of 1–4 nm (revised Suppl. Fig. 9a, b). These results are consistent with the observation from the DSLR photos, which indicate the OM film kept $< 3 \text{ nm}$ colour shift as the curvature increases to 0.5 mm^{-1} (revised Suppl. Fig. 9c–g). In addition, after releasing the bending and placing the OM film back onto silicon wafer, the reflectance spectrum remains almost unchanged in comparison to its pristine status (revised Suppl. Fig. 9h). Thus we conclude that our bending conditions did not permanently deform the OM structure under this condition.

Eventually there will be a point where the stress/strain is enough to deform the structure and change the spectrum permanently and this depends on a number of factors including the molecular weight of the polymer. The maximum curvature tested was 0.5 mm^{-1} (radius of curvature: 2 mm). With the OM film thickness being 1–2 μm and the aluminium foil thickness being 60 μm , the maximum strain applied is estimated as 1.5% (strain = distance from neutral axis / radius of curvature). We expect that more extreme bending is required to achieve the required strain for deforming the OM structure. The study of such extreme bending conditions are not the major purpose of this paper, thus we would like to report those studies in a separate article in future.

In the revised manuscript, we clearly specify the experimental conditions on our bending statements. The revisions made in the main text and Suppl. Info. provide further insights, which are appended as follows:

Main text, page 8, line 169–175, the description about mechanical bending experiments is revised as follows:

“OM films made by polycarbonate were transferred onto the aluminium foil (60- μm thickness) to study the effect of bending on the structural colour. The in situ spectrum shows negligible colour change when the curvature of the film is $\leq 0.1 \text{ mm}^{-1}$. As the curvature further increases to 0.5 mm^{-1} , the Bragg peak shift of OM film remains in the small range of 1–4 nm. The spectrum remains almost unchanged after releasing the bending and returning the film to silicon wafer (see Supplementary Fig. 9 for details).”

Suppl. Info., page 10–11, a revised Supplementary Fig. 9 is presented (please see the next page):

Supplementary Fig. 9 | Impact of bending on the structural colour of OM film. **a**, In situ reflectance spectrum of the OM film during the bending test. The spectrum reference was aluminium (LUXAL UV, 60- μm -thick, Toyo Aluminium K.K.). **b**, Bragg peak shift versus the curvatures of OM film. **c, d**, Side-view and top-view photos of an OM film on the aluminium foil before the bending test. **e, f**, Side-view and top-view photos of the film during the bending test. **g**, Hue and colour shift versus the curvatures of OM film. **h**, In the end, bending was released and the OM film was placed back onto silicon wafer for spectral comparison. Scale bar in (c) and (e), 10 mm. The OM films were made using Polycarbonate (PC)/4,4'-bis-(diethylamino)-benzophenone (BDABP) on silicon wafer (LED $\lambda_i = 385$ nm). The film was peeled off from the substrate carefully in water and transferred onto a piece of aluminium foil. A vice was used for the bending to apply various curvatures to the OM film. As the bending was increased, the in situ

reflectance spectrum was recorded. Side-view photos were taken at the same time to obtain the curvature. The curvature was calculated by the software *Fiji* (version 2.1.0/1.53c) using the built-in plugin *Kappa*. Reported values are the average curvature of a 2.5-mm-length curve at the apex region of the bent film where its reflectance spectrum was recorded. In situ measurements show little spectral change when the curvature of OM film is $\leq 0.1 \text{ mm}^{-1}$ (radius of curvature: $\geq 10 \text{ mm}$). As the curvature further increases to 0.5 mm^{-1} (radius of curvature decreases to 2 mm), the Bragg peak shift of OM film remains in the small range of 1–4 nm.

As an alternative approach to check the impact of bending, top-view photos were taken for colour comparison under different bending conditions. To keep consistency, an 18% neutral grey card was used as the colour reference. Colour comparison focused on the apex region of the bent OM film that held a fixed angle to the camera lens. The wavelength value of the colour is derived from the hue value according to the visible light spectrum chart, using *ImageJ* software (version 1.52p). Photo comparison indicates that the colour shift of OM film keeps $< 3 \text{ nm}$ as its curvature increases to 0.5 mm^{-1} . This is consistent with the results from the spectral measurements.

The results show negligible change of structural colour in this range of bending conditions. In addition, after releasing the bending and placing the OM film back onto silicon wafer, the reflectance spectrum remains almost unchanged in comparison to its pristine status. This indicates bending to such an extent does not permanently deform the OM structure.

REVIEWERS' COMMENTS

Reviewer #3 (Remarks to the Author):

After revision, the authors did quite a few works to improve the quality of paper. I would recommend this manuscript for this format to be accepted and published.